# Awake responses suggest inefficient dense coding in the mouse retina

**Tom Boissonnet[1,2†], Matteo Tripodi[1], Hiroki Asari[1]***

[1]Epigenetics and Neurobiology Unit, EMBL Rome, European Molecular Biology Laboratory, Monterotondo, Italy; [2]Collaboration for joint PhD degree between EMBL and Université Grenoble Alpes, Grenoble Institut des Neurosciences, La Tronche, France

**Abstract** The structure and function of the vertebrate retina have been extensively studied across species with an isolated, ex vivo preparation. Retinal function in vivo, however, remains elusive, especially in awake animals. Here, we performed single-unit extracellular recordings in the optic tract of head-fixed mice to compare the output of awake, anesthetized, and ex vivo retinas. While the visual response properties were overall similar across conditions, we found that awake retinal output had in general (1) faster kinetics with less variability in the response latencies; (2) a larger dynamic range; and (3) higher firing activity, by ~20 Hz on average, for both baseline and visually evoked responses. Our modeling analyses further showed that such awake response patterns convey comparable total information but less efficiently, and allow for a linear population decoder to perform significantly better than the anesthetized or ex vivo responses. These results highlight distinct retinal behavior in awake states, in particular suggesting that the retina employs dense coding in vivo, rather than sparse efficient coding as has been often assumed from ex vivo studies.

**\*For correspondence:**
asari@embl.it

**Present address:** [†]Center for Advanced Imaging, Heinrich-Heine-Universität Düsseldorf, Düsseldorf, Germany

**Competing interest:** The authors declare that no competing interests exist.

## Editor's evaluation

This paper compares output signals from the mouse retina in three conditions: awake mice, anesthetized mice, and isolated retinas. The paper provides compelling evidence for substantial and important differences, particularly between awake and either of the other conditions. Retinal signaling has been well studied using ex vivo preparations, with an assumption that the findings from those studies can be carried over to how the retina operates in vivo. The results from this paper at a minimum indicate a need to be cautious about that assumption.

## Introduction

The vertebrate retina is one of the best characterized parts of the central nervous system (*Gollisch and Meister, 2010*; *Masland, 2012*). It consists of ~100 cell types in total (*Shekhar et al., 2016*; *Yan et al., 2020*), including ~30 types of retinal ganglion cells (RGCs) that send the retinal output signals to the brain via the optic nerve (*Sanes and Masland, 2015*; *Baden et al., 2016*). Each of these RGC types forms distinct neural circuits within the retina to extract specific features of the visual image coming into the eye, such as color, contrast, and motion. The retina thus performs parallel and dynamic processing as the first stage of the visual pathway.

Most of our knowledge on retinal function has been obtained from ex vivo studies because isolated retinal tissues are nevertheless functional, for example, responsive to light (*Barlow et al., 1964*). While powerful, however, ex vivo physiological approaches have certain limitations. First, one cannot perform long-term recordings (*Meister et al., 1994*). Second, one cannot avoid artifacts due to retinal dissection, such as the effect of cutting RGC axons (*Vidal-Sanz et al., 2017*) or retinal epithelial

**eLife digest** When light enters the eyes, it is focused onto the retina, a thin layer of brain tissue at the back of the eye. The retina converts light information into electrical signals that are transmitted to the rest of the brain to perceive vision. Unlike the rest of the brain, this light-processing tissue can continue working even when removed from an animal, making it easier for scientists to study how the retina works. This has helped it become one of the best-understood parts of the brain.

Most knowledge of retinal signal processing comes from studies of isolated retinas. However, it was still unclear if these samples behave the same way as they do in live animals, and whether findings in isolated retinas apply to natural visual processing in an awake state.

To determine this, Boissonnet et al. compared the visual responses of the retina in awake mice, anesthetised mice and when isolated from mice. Measurements of retinal electrical signals showed that awake mice responded to light substantially more quickly and strongly than the others. Computational analysis suggested that the amount of information carried to the brain was largely comparable across the different subjects, but the retina in awake mice used more energy.

The findings indicate that further studies are needed to better understand how the retina processes visual information in awake animals, rather than just in isolated conditions. Progressing this understanding could ultimately help to develop prosthetic devices that can act as a retina in the future.

detachment (*Strauss, 2005*). Neuromodulatory effects of retinopetal pathways are also difficult to study in an isolated retinal preparation (*Repérant et al., 2006*; *Esposti et al., 2013*). In vivo studies are thus indispensable for understanding to what degrees our knowledge on the retina can be translated from ex vivo to fully physiological in vivo conditions and clarifying retinal function thoroughly.

Previous physiological studies on the retina in vivo were conducted mostly under anesthetized— and often paralyzed—conditions. To monitor RGC activity in vivo, for example, single-unit recordings were made directly from the retina (*Kuffler, 1953*), at the optic nerve or tract fibers (*Hartline, 1938*; *Lettvin et al., 1959*; *Enroth-Cugell and Robson, 1966*; *Mastronarde, 1983*; *Mastronarde, 1985*; *Sagdullaev and McCall, 2005*), or in the form of 'slow-potential' in the dorsal lateral geniculate nucleus (dLGN; *Bishop et al., 1962*; *Kaplan and Shapley, 1984*). Optical methods were also developed to image the activity of retinal neurons directly through the pupil of a live animal under anesthesia (*Geng et al., 2012*; *Yin et al., 2013*; *Yin et al., 2014*). In contrast, thus far only a handful of studies have reported awake recordings from the retina (*Esposti et al., 2013*; *Hong et al., 2018*) or its outputs (*Weyand, 2007*; *Liang et al., 2018*; *Liang et al., 2020*; *Schröder et al., 2020*; *Sibille et al., 2022*). Thus, despite a long history of research on the retina, it still remains unclear what exactly the eye tells the brain in awake animals.

As recordings from awake behaving animals became routine for many brain areas (*Dombeck et al., 2007*; *Jun et al., 2017*), growing attention has been paid to the roles of an animal's behavior and internal brain states in the function of the sensory systems (*Niell and Stryker, 2010*; *Lee and Dan, 2012*). For example, systematic studies on the early visual pathway showed higher firing activity and faster response dynamics in both dLGN (*Durand et al., 2016*) and the superior colliculus (SC; *De Franceschi and Solomon, 2018*) of awake animals than those under anesthesia. A critical question that was left unanswered is to what extent such differences originate in the retina.

To clarify differences in the retinal visual response properties between awake and anesthetized conditions, here we employed single-unit extracellular recording techniques from head-fixed mice. Specifically, the electrodes were placed in the optic tract (OT), a bundle of nerve fibers composed of RGC axons as they project from the eye to their main targets: dLGN and SC (*Ellis et al., 2016*; *Román Rosón et al., 2019*). These recordings are superior to direct in vivo retinal recordings (e.g., with epiretinally implanted mesh electrodes; *Hong et al., 2018*) because the retinal circuits and the eye optics remain intact, and also to those extracellular recordings in SC (*Sibille et al., 2022*) or dLGN (*Weyand, 2007*) because RGC axonal signals do not need to be disambiguated from those of local axons or somata. This advantage also exists for calcium imaging recordings of the retina directly in the eye of immobilized zebrafish larvae (*Esposti et al., 2013*) or those of RGC axon terminals in dLGN (*Liang et al., 2018*; *Liang et al., 2020*) or SC (*Schröder et al., 2020*; *Molotkov et al., 2022*). These imaging approaches, however, lack the temporal precision as in the electrophysiology,

which is an important aspect of the information processing in the retina (*Gollisch and Meister, 2008*).

From our OT recordings, we examined the visual responses to a set of visual stimuli widely used for probing retinal function ex vivo (*Baden et al., 2016*; *Jouty et al., 2018*), including moving gratings, white-noise stimuli, and full-field flickering stimuli at different temporal frequencies. We used two different anesthetics that are commonly used in neuroscience research: isoflurane gas and an intraperitoneal combination of fentanyl, medetomidine, and midazolam (FMM). In both cases, we found that the temporal dynamics of the retinal outputs were slower than in awake recordings, consistent with the previous studies on the effects of anesthesia in retinorecipient areas (dLGN, *Durand et al., 2016*; SC, *De Franceschi and Solomon, 2018*). We also found that the retinal outputs in an awake condition had higher baseline firing rates and a larger dynamic range than in anesthetized or ex vivo conditions. Consequently, while information was conveyed at a comparable rate in bits per second, the amount of information per spike was lower in the awake responses. Furthermore, we found that a linear decoder performed significantly better with awake population responses. Thus, our results do not support sparse efficient coding (*Attneave, 1954*; *Barlow and Rosenblith, 1961*) or minimum energy principles (*Laughlin, 2001*) that have been used as a model of the early visual processing (*Atick and Redlich, 1990*; *Gjorgjieva et al., 2019*; but see *Schwartz, 2021*). We instead suggest that the retina may employ dense coding principles in vivo.

## Results
### Characterization of retinal output responses in vivo

To monitor retinal output in vivo, we established extracellular single-unit recording methods from axons of RGCs in the OT of head-fixed mice (*Figure 1*). In total, we made 17 chronic (75 cells) and 80 acute recordings (207 cells without anesthesia; 325 cells with isoflurane; 103 cells with FMM), where a standardized set of visual stimuli were presented to the subject animal to characterize the visual response properties of the recorded cells (for ~1 hr; *Figure 1A, B*; see Methods for details). Isolated units typically had a triphasic spike waveform (e.g., *Figure 1F–H*) as expected from axonal signals (*Barry, 2015*) and most of them had a minimum interspike interval above 2 ms (e.g., *Figure 1I–K*; 93%, awake; 94%, isoflurane; 99%, FMM). Histological verification of the electrode location (e.g., *Figure 1E*) further ensured a successful OT recording in vivo.

We performed physiological classification of the recorded cells (*Figure 2*; see Methods for details). Specifically, we first identified reliably responsive cells based on the visual responses to full-field contrast-inverting stimuli (*Figure 2A, B*), and categorized their response polarity into ON, ON/OFF, and OFF types (*Figure 2C*). In both anesthetized and awake conditions, we found around 20–50% of ON cells, 10–40% of ON/OFF cells, and 20% of OFF cells in our data sets. We then further classified the cells from the viewpoint of orientation selectivity (OS) and motion-direction selectivity (DS), based on the responses to moving gratings in eight different directions (*Figure 2D–F*). In the anesthetized conditions, about a half of the cells (42–54%) were either OS or DS or both regardless of their response polarities or anesthetics. This is consistent with a previous in vivo study (*Hong et al., 2018*), but rather higher than one would expect from anatomical and physiological studies ex vivo (about a third; *Sanes and Masland, 2015*; *Baden et al., 2016*; *Jouty et al., 2018*). In contrast, we found much less DS/OS cells (17%) in the awake condition. This discrepancy in the fraction of DS/OS cells cannot be explained by a sampling bias per se, if any, because we used the same experimental setup between the anesthetized and awake recordings. Instead, it likely arose from a difference in the baseline firing rate (see below and *Figure 3*). Since DS/OS index is defined as a normalized population vector (*Equation 3* in Methods; *Mazurek et al., 2014*), higher firing in the awake condition generally leads to a larger denominator, hence a lower DS/OS index value.

A distinguishing feature of awake recordings is an ongoing behavior of the animal during recordings (*Figure 1—figure supplement 1*). Behavioral states affect sensory processing (*Niell and Stryker, 2010*), even at the early stages including the retina (*Schröder et al., 2020*). This drove us to examine the effects of eye movements, pupil dynamics, as well as locomotion on the retinal output in vivo.

First, we found that eye movements were not uniformly distributed in all directions (*Figure 1— figure supplement 1H*). Specifically, saccades were primarily in the horizontal directions and much less in the vertical directions. Nevertheless, as reported previously (*Samonds et al., 2018*; *Miura and*

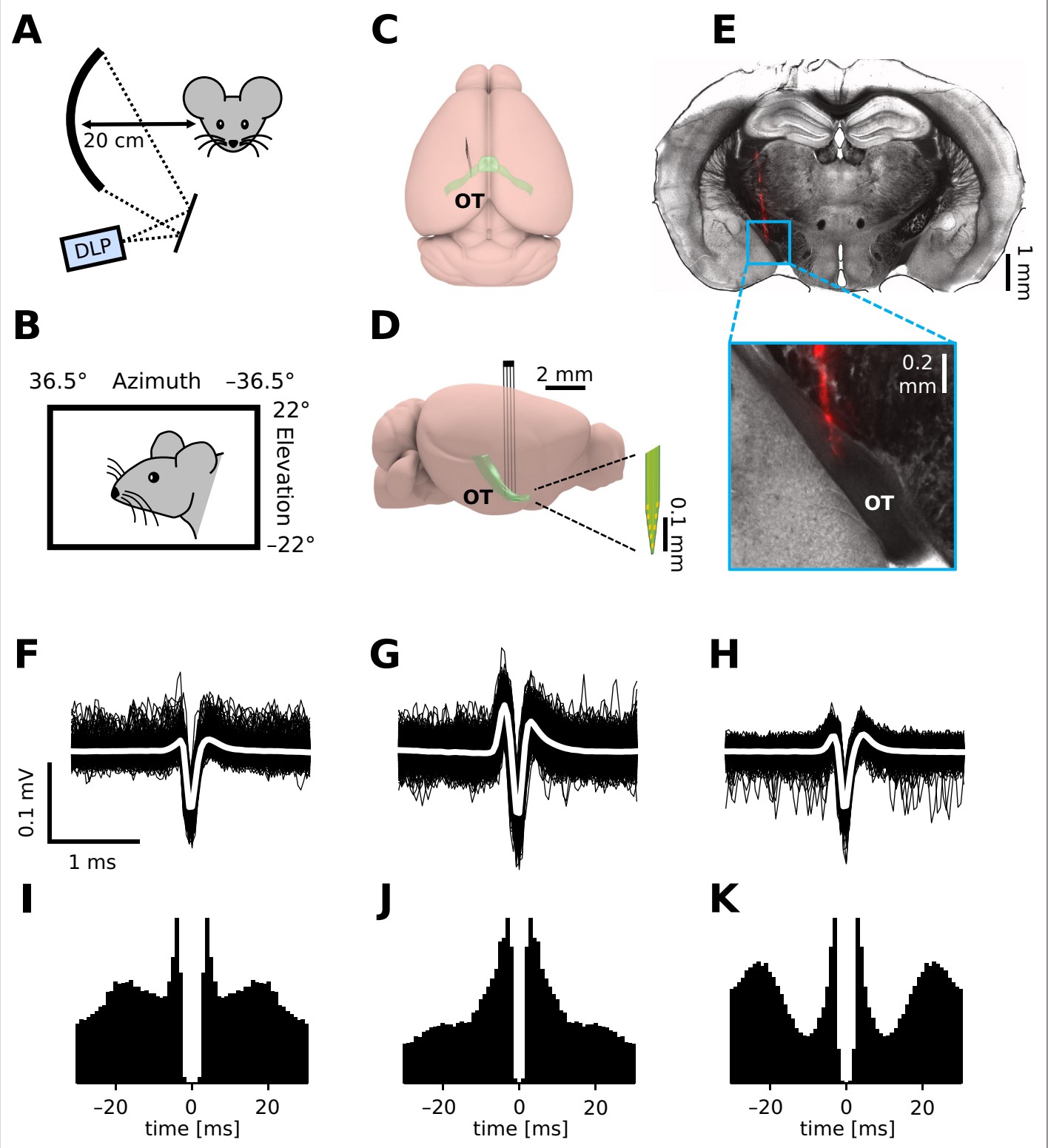

**Figure 1.** In vivo extracellular recordings from the mouse optic tract (OT). (**A, B**) Schematic diagram of the experimental setup. We presented visual stimuli to a head-fixed mouse using a digital light processing (DLP) device projecting images onto a spherical screen placed laterally to the subject animal (**A**, front-view; **B**, side-view). See Methods for details and specifications. Schematic diagram (**C**, top-view; **D**, side-view) of the brain and electrode location to target the OT. (**E**) Histological image of a representative brain sample (coronal section, 150 μm thick) showing the electrode trace (red,

*Figure 1 continued on next page*

*Figure 1 continued*

DiI stain deposited on the electrode). Spike waveform (**F–H**; black, individual trials; white, mean) and autocorrelogram (**I–K**; bin size, 1 ms) for three representative units recorded from the OT of an awake mouse.

The online version of this article includes the following figure supplement(s) for figure 1:

**Figure supplement 1.** Eye motion and behavioral data analysis.

*Scanziani, 2022*), head-fixed awake mice did not frequently make eye movements (0.06 ± 0.03 Hz; mean ± standard deviation, 19 animals) and thus their effects on RGC activity were considered to be negligible in this study. The effects of microsaccades were also negligible here as we mainly used full-field stimulation.

We next examined the relationship between pupil size and the retinal output dynamics. They were both dependent on the light intensity level as evident by the pupillary reflexes (*Hattar et al., 2003*) and the visual response polarities (*Figure 2C*), respectively. To analyze their relationship, we thus focused on the recording period where the light intensity was relatively stable over time (see Methods), and found that pupil size was correlated with RGC firing in a response-polarity-dependent manner (*Figure 2—figure supplement 1A, B*; Pearson's $R = 0.38$, $p < 0.001$), likely mediated by the amount of light passing through the pupil.

Lastly, as reported previously (*Schröder et al., 2020*), we found that the activity of some cells was significantly correlated with an animal's locomotion speed (*Figure 2—figure supplement 1C, D*). While there was a positive correlation between pupil size and locomotion speed (0.34 ± 0.20 from 19 animals; see also *Erisken et al., 2014*), unlike the pupil size effects, the locomotion effects on the retinal output in vivo was not dependent on the visual response polarity of the cells (*Figure 2—figure supplement 1D*; Pearson's $R = 0.11$, $p = 0.3$). Thus, the correlation between locomotion and RGC firing dynamics cannot be simply explained by the light intensity fluctuation due to the associated pupil size dynamics.

## Higher firing in awake than in anesthetized mice

For characterizing the retinal output properties in vivo, we first analyzed the responses to full-field sinusoidally flickering stimuli that linearly changed the amplitude over time (from 0 to 100% contrast for 10 s; see *Figure 3A–C* for representative responses). We fitted to the response an even power of sine function, weighted with a sigmoid envelope (*Equation 6* in Methods), and examined the curve fit parameter values to analyze the response properties in different recording conditions.

First, we found significantly higher baseline firing rates in awake mice ($B = 20 ± 27$ Hz; median ± interquartile range) than in those under anesthesia (isoflurane, 2 ± 6 Hz, $p < 0.001$, *U*-test; FMM, 2 ± 4 Hz, $p < 0.001$; *Figure 3D*). Awake ON cells had particularly high baseline activity (up to ~100 Hz) and showed a prominent reduction in firing rates in response to light decrements (see *Figure 3B* for example). In contrast, due to this high baseline activity, many of them showed virtually no responses to light increments ($N = 36$ out of 84), except for a sharp rebound response to a full-contrast inversion ($N = 26$ out of 36; see *Figures 2A and 3B* for example). These ON cells had negative amplitudes in the curve fit (*Figure 3E*, green) with the same response phase as OFF cells (*Figure 3F*; see *Figure 3A* for a representative OFF cell's responses). This indicates that they primarily encode light decrements by decreasing firing from their high baseline firing rates (62 ± 32 Hz, mean ± standard deviation), rather than light increments by increasing firing as conventional ON cells do (*Masland, 2012*; *Sanes and Masland, 2015*; *Baden et al., 2016*; *Jouty et al., 2018*). We thus categorized these cells independently as an 'OFF-suppressive' class (*Figure 2G*). This is likely an emerging property of the cells due to high baseline firing in awake mice. Indeed, such responses were barely observed in anesthetized animals (isoflurane, 2/325 cells; FMM, 1/103 cells; *Figure 2H1*) where the baseline firing rate was generally low (*Figure 3D*). As expected from previous ex vivo studies (*Masland, 2012*; *Sanes and Masland, 2015*; *Baden et al., 2016*; *Jouty et al., 2018*), anesthetized ON cells increased firing upon light increments (see *Figure 3C* for a representative response). Positive amplitudes in the curve fit (*Figure 3E*) with an opposite response phase from OFF cells (*Figure 3F*) further support that these ON cells encode light increments. These data suggest that the retinal coding depends on the baseline firing level of RGCs.

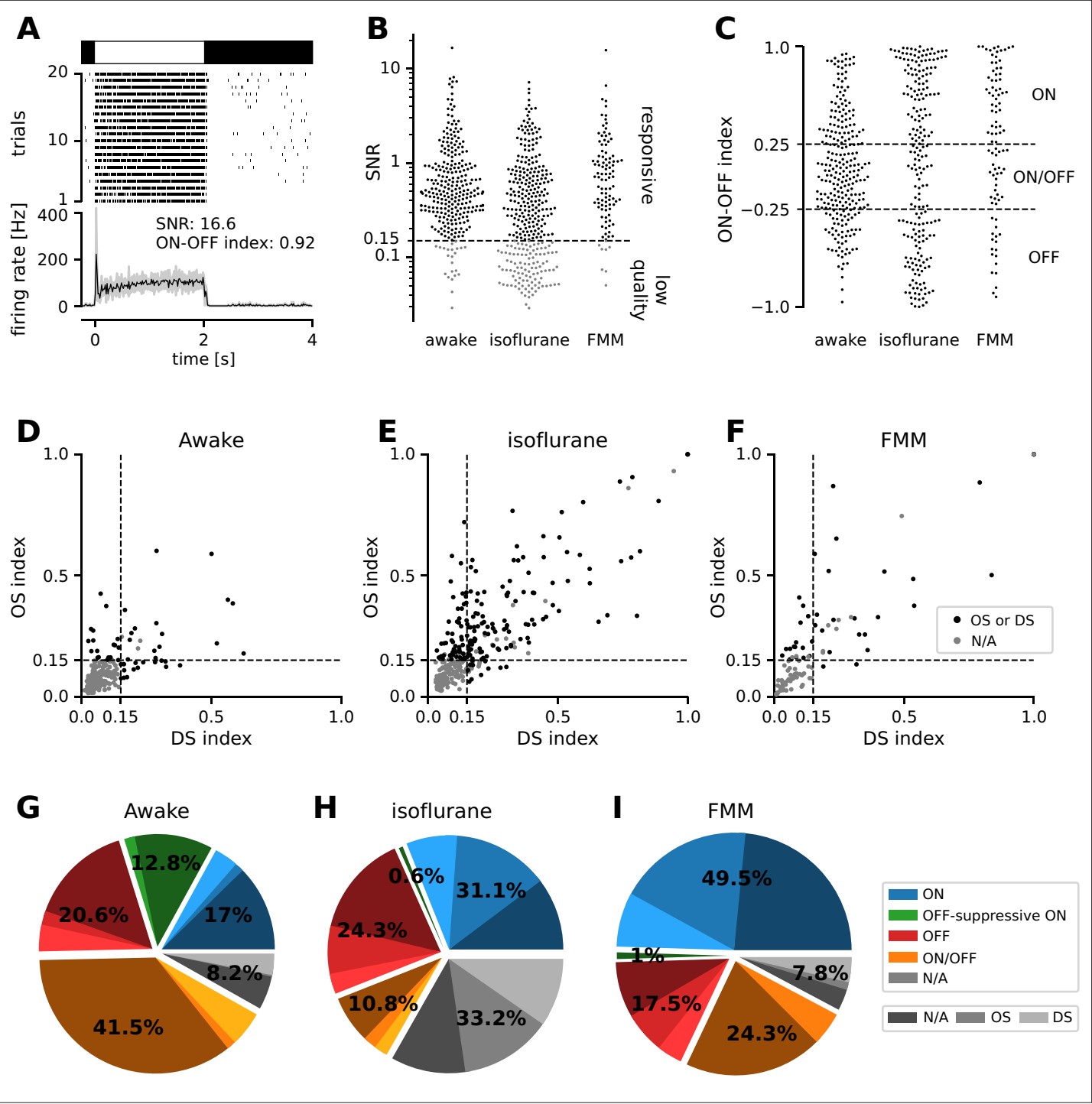

**Figure 2.** Physiological classification of retinal output responses in vivo. (**A**) Representative retinal output responses to full-field contrast-inverting stimuli: top, stimulus; middle, raster graph over trials; bottom, peri-stimulus time histogram (black, mean; gray, variance; signal-to-noise ratio [SNR], *Equation 1* in Methods; ON–OFF index, *Equation 2*). (**B**) SNR of the retinal output responses in different recording conditions. We set a threshold at 0.15 to identify reliably responsive cells (black) and low-quality unclassifiable cells (gray). (**C**) ON–OFF index distributions from the reliably responsive cells. While no apparent clusters were identified, we set a threshold at ±0.25 to categorize the response polarity into ON, ON/OFF, and OFF cells. Within the ON cells, we further identified those with an 'OFF-suppressive' response to the full-field flickering stimulus (*Figure 3*). Distribution of direction selectivity (DS)/orientation selectivity (OS) indices (*Equation 3*) in each recording condition (**D**, awake; **E**, isoflurane; **F**, fentanyl, medetomidine, and midazolam [FMM]). We set a threshold at 0.15 (with p < 0.2) to identify whether cells are OS/DS (black) or not (gray). (**G–I**) Fraction of identified response classes in vivo: ON (blue; OFF-suppressive in green), OFF (red), ON/OFF (orange), and the rest unclassifiable cells ('N/A', gray). Cells in each

*Figure 2 continued on next page*

*Figure 2 continued*

category were further divided based on the OS/DS properties (hued). The OFF-suppressive ON cells were prominent in the awake condition (**G**, 36/282 cells), but rarely observed under anesthesia (**H**, isoflurane, 2/325 cells; **I**, FMM, 1/103 cells).

The online version of this article includes the following figure supplement(s) for figure 2:

**Figure supplement 1.** Retinal output was correlated with an animal's behavioral states.

Second, we found that the awake responses had a larger dynamic range than those under anesthesia. Specifically, the absolute amplitude of the evoked responses from the baseline was larger for the awake condition (*Figure 3E*; awake, |A| = 45 ± 55 Hz, median ± interquartile range; isoflurane, 30 ± 31 Hz, $p < 0.001$, Mann–Whitney *U*-test against awake data; FMM, 27 ± 33 Hz; $p < 0.001$). Moreover, the sigmoid envelope had a higher midpoint (awake, $t_0$ = 70 ± 61% contrast; isoflurane, 45 ± 32% contrast, $p < 0.001$; FMM, 45 ± 44% contrast, $p < 0.001$) and a lower steepness (awake, $\lambda$ = 0.44 ± 0.53/s; isoflurane, 0.67 ± 0.79/s, $p < 0.001$; FMM, 0.69 ± 0.62/s, $p < 0.001$) for the awake responses than for the anesthetized ones.

We next examined the sensitivity to contrast. At low stimulus contrast, we found a higher sensitivity in the awake condition (estimated evoked response at 10% contrast = 7 ± 13 Hz, median ± interquartile range) than under anesthesia (isoflurane, 4 ± 12 Hz, $p < 0.001$, Mann–Whitney *U*-test; FMM, 3 ± 10 Hz, $p < 0.001$). However, the maximum sensitivity to a change in contrast—that is, the slope size

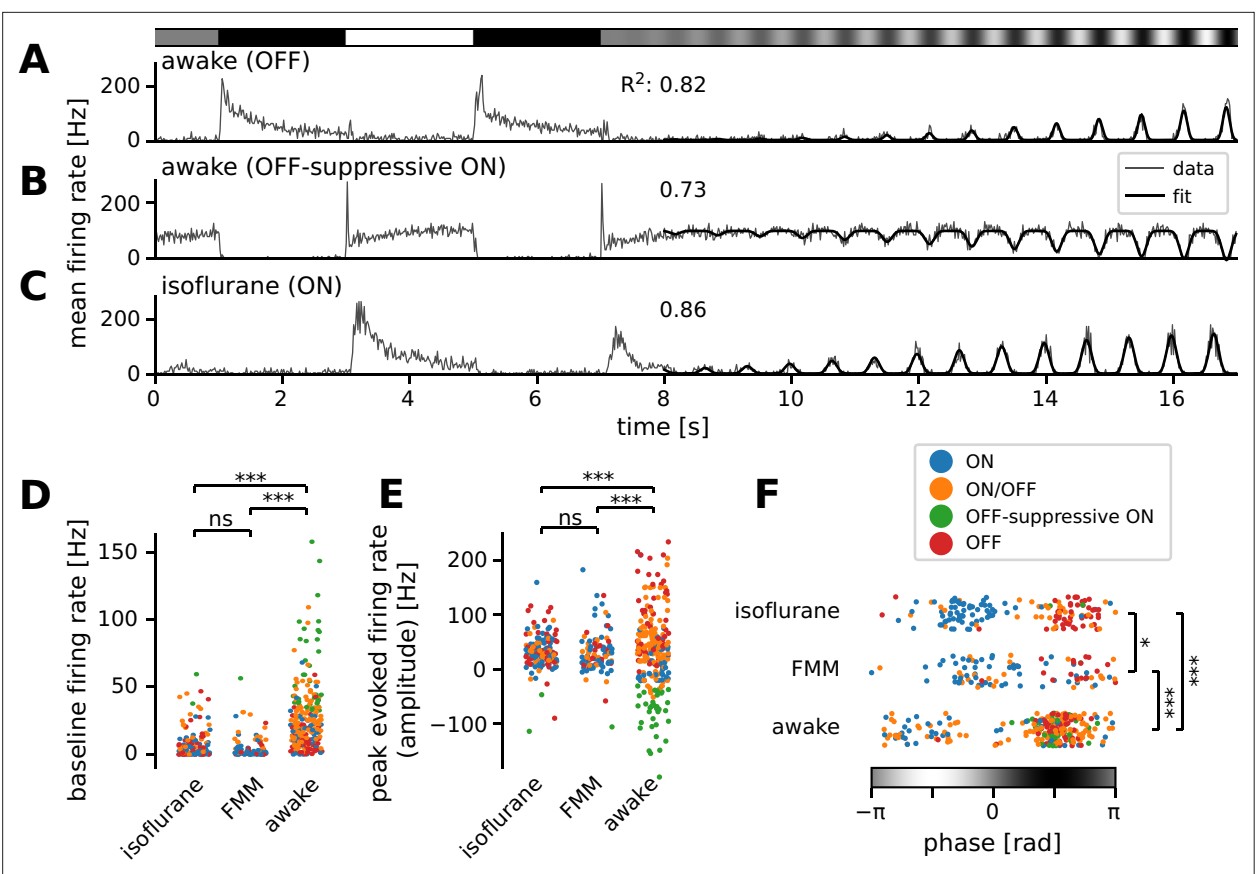

**Figure 3.** Many ON retinal ganglion cells showed suppressive OFF responses in awake condition. Mean firing rate of representative cells in response to a sinusoidally flickering stimulus with increasing contrast in the awake (**A**, OFF; **B**, OFF-suppressive ON, the same cell as in *Figure 2A*) or anesthetized conditions (**C**, ON). Overlaid with the peri-stimulus time histogram (gray) is the model fit (black, *Equation 6* in Methods). The number on top indicates the fit quality (explained variance $R^2$ in *Equation 5*). Population data of the model parameters (**D**, baseline *B*; **E**, amplitude *A*; **F**, phase *ϕ*) across different conditions: isoflurane anesthesia (N = 147), fentanyl, medetomidine, and midazolam (FMM) anesthesia (N = 95), and awake (N = 247 in total). The sinusoidal stimulus pattern relative to the response peak is also indicated at the bottom of F. Cell types are color coded as in *Figure 2* (blue, ON; green, OFF-suppressive ON; red, OFF; orange, ON/OFF). Note high baseline with negative amplitude and positive phase for the OFF-suppressive ON cells, which were predominantly found in the awake condition: \*\*\*p < 0.001; \*p < 0.05; ns, nonsignificant (**D**, *U*-test; **E**, *U*-test on the absolute values; **F**, *t*-test).

at the midpoint of the sigmoid—was comparable between the awake and anesthetized conditions ($|A\lambda/4| = 5 \pm 11$ Hz/%contrast; isoflurane, $6 \pm 9$ Hz/%contrast; FMM, $5 \pm 7$ Hz/%contrast; $p = 0.1$, Kruskal–Wallis test).

We also found a significant phase shift in the responses between the recording conditions (*Figure 3F*). This phase parameter in the curve fit represents the position of the response peak relative to the sinusoidal stimulus intensity patterns (see Methods for details), and we identified two clusters in each data set: the one with negative phase for ON cells (*Figure 3F*, blue) and the other with positive phase for OFF and OFF-suppressive ON cells (*Figure 3F*, red and green, respectively). ON/OFF cells were found in either cluster, depending on the relative strength of their responses to light increments versus light decrements (*Figure 3F*, orange). In both clusters, we found that the phase was on average smallest for the awake condition ($-1.9 \pm 0.6$ and $1.7 \pm 0.6$ radian, respectively; mean ± standard deviation; isoflurane, $-0.9 \pm 0.6$ and $2.0 \pm 0.5$ radian; $p < 0.001$ for both cases, $t$-test), while largest for the FMM anesthesia ($-0.5 \pm 0.8$ and $2.3 \pm 0.5$ radian; $p = 0.007$ and $p = 0.044$ against corresponding isoflurane data, respectively). This implies that the response dynamics are faster in awake mice than those under anesthesia, especially with FMM.

Finally, we analyzed the retinal output properties between different visual response polarities by combining data across recording conditions (ON, 199 cells in total; OFF, 121 cells; and ON/OFF, 170

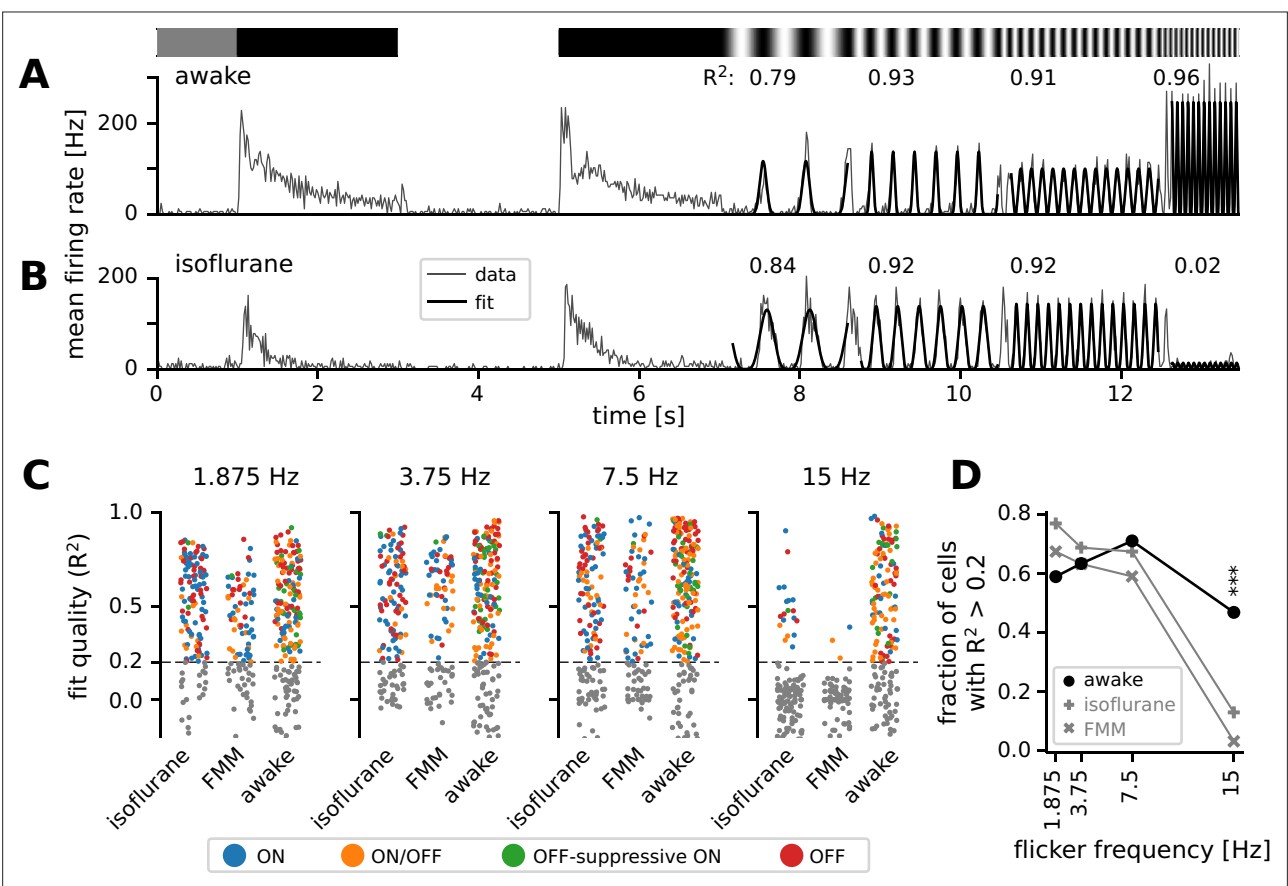

**Figure 4.** Retinal output showed higher temporal frequency sensitivity in awake than in anesthetized mice. (**A**) Representative retinal output (gray, mean firing rate over 10 trials) in an awake condition in response to full-field sinusoidally flickering stimuli at different temporal frequencies (1.875, 3.75, 7.5, and 15 Hz), following full-field contrast inversions. Overlaid is the curve fit (*Equation 4* in Methods; black). The number on top is the explained variance of the curve fit ($R^2$, *Equation 5* in Methods), representing the fit quality. (**B**) Representative retinal output responses under isoflurane anesthesia (shown in the same format as in A). (**C**) Population data of the fit quality at four different stimulus frequencies in the awake ($N = 248$) or anesthetized conditions (isoflurane, $N = 147$; fentanyl, medetomidine, and midazolam [FMM], $N = 95$), color coded for the responses class as in *Figure 2*. The fit quality threshold was set to be 0.2 (black, $R^2 \geq 0.2$; gray, $R^2 < 0.2$). (**D**) Fraction of the cells with the fit quality above the threshold across different conditions (awake, black line with circles; isoflurane, gray line with vertical crosses; FMM, gray line with diagonal crosses), representing the frequency tuning of the retinal output at the population level. A significantly larger fraction of cells was responsive at 15 Hz in the awake condition than in the anesthetized conditions (***$p < 0.001$ for both isoflurane and FMM; two-proportion $z$-test).

cells). We found that the baseline firing rate was higher for ON/OFF cells (17 ± 24 Hz; median ± inter-quartile range) than for ON or OFF cells (5 ± 24 and 6 ± 10 Hz, respectively; $p < 0.001$, Kruskal–Wallis test). Otherwise we did not find any statistically significant dependency on the response polarity at the population level.

## Faster response kinetics in awake than in anesthetized mice

To better characterize the kinetics of retinal output in vivo, we next analyzed the responses to full-field sinusoidal flickers at different temporal frequencies (*Figure 4*). In awake mice, most cells responded faithfully to all the stimulation frequencies we tested from 1.875 to 15 Hz (see *Figure 4A* for example). Some awake cells showed the largest responses to the 15 Hz stimulation (*Figure 4A*; curve fit quality with *Equation 4*), suggesting that they were possibly tuned to even higher frequencies. In the anes-thetized conditions, in contrast, retinal output responses typically followed the stimulation frequen-cies up to 7.5 Hz, but failed to do so at 15 Hz (see *Figure 4B* for example). For quantification, we fitted an even power of sine function to the responses (*Equation 4* in Methods) and set a threshold of the curve fit quality (defined as the explained variance; *Equation 5*) at 0.2 to select cells with robust responses (*Figure 4C, D*). In the awake condition, a larger number of cells responded well at medium frequencies (3.75 Hz, 63%; 7.5 Hz, 71%) than at a low frequency (1.875 Hz, 59%); and the majority of the awake cells remained responsive to a high-frequency stimulus (15 Hz, 47%). Under anesthesia, in contrast, the fraction of responsive cells was the largest at a low frequency (1.875 Hz; isoflurane, 77%; FMM, 67%). Fewer cells responded robustly at faster flicker rates (3.75 and 7.5 Hz; isoflurane, 69% and 67%, respectively; FMM, 63% and 59%), and only a small fraction of the anesthetized cells were able to follow the stimulation frequency at 15 Hz (isoflurane, 13%; FMM, 3%). Consistent with the analysis of the response phase (*Figure 3*), these outcomes indicate that the retinal output dynamics are faster in awake animals than those under anesthesia, especially for FMM.

Across the response polarities, we found that ON/OFF cells ($N$ = 170 in total) were most responsive at a medium frequency (7.5 Hz, 59%), while ON or OFF cells ($N$ = 199 and 121, respectively) were generally responsive from low to medium frequencies (1.875–7.5 Hz; 68–73% and 79–82%, respec-tively). Indeed, the fraction of responsive cells under our criteria was significantly lower for ON/OFF cells (1.875 Hz, 46%; 3.75 Hz, 49%) than for ON or OFF cells at low frequencies ($p < 0.001$ for both frequencies, two-proportion $z$-test). No dependency on the response polarity was found for a high-frequency stimulation (15 Hz; ON, 25%; OFF, 29%; ON/OFF, 31%).

## Comparison of retinal output properties between in vivo and ex vivo

We have thus far focused on the retinal output properties in vivo, and showed higher baseline firing rates (*Figure 3*) and faster response kinetics (*Figure 4*) in awake mice than in those under anesthesia. We next asked if RGC responses differ between in vivo and ex vivo conditions. This is a critical compar-ison because retinal physiology has been mostly studied and best characterized in an isolated prepa-ration (*Gollisch and Meister, 2010*; *Sanes and Masland, 2015*), but little is known about it in awake animals (*Weyand, 2007*; *Liang et al., 2018*; *Liang et al., 2020*; *Schröder et al., 2020*; *Sibille et al., 2022*). To what extent can we translate our knowledge on the retina from ex vivo to in vivo? Here, we exploited stimulus ensemble statistical techniques ('reverse correlation'; *Meister et al., 1994*; *Chichilnisky, 2001*) to systematically characterize the visual response properties and make a direct comparison across different recording conditions in the linear–nonlinear cascade model framework (see Methods for details). Specifically, using full-field white-noise stimuli, we analyzed (1) the linear temporal filter (*Figure 5*), estimated by the spike-triggered average (STA) stimulus, that is, the mean stimulus that triggered spiking responses; and (2) the static nonlinear gain function (*Figure 6*), that is, an instantaneous mapping of the STA output to the neural responses (*Equation 7* in Methods). Here we reanalyzed the existing data sets for ex vivo recordings (696 cells from 18 isolated mouse retinas recorded at 37°C; *Vlasiuk and Asari, 2021*).

## Faster response kinetics in awake condition than ex vivo

We identified a good quality STA in more than two-thirds of the cells recorded in vivo (e.g., *Figure 5A*). To compare the temporal dynamics of the STAs across different recording conditions, we used the following two measures: (1) the first peak latency of the STA estimated from a difference-of-Gaussian curve fit (*Figure 5A*); and (2) spectral peak frequency calculated by the Fourier transform of the fitted

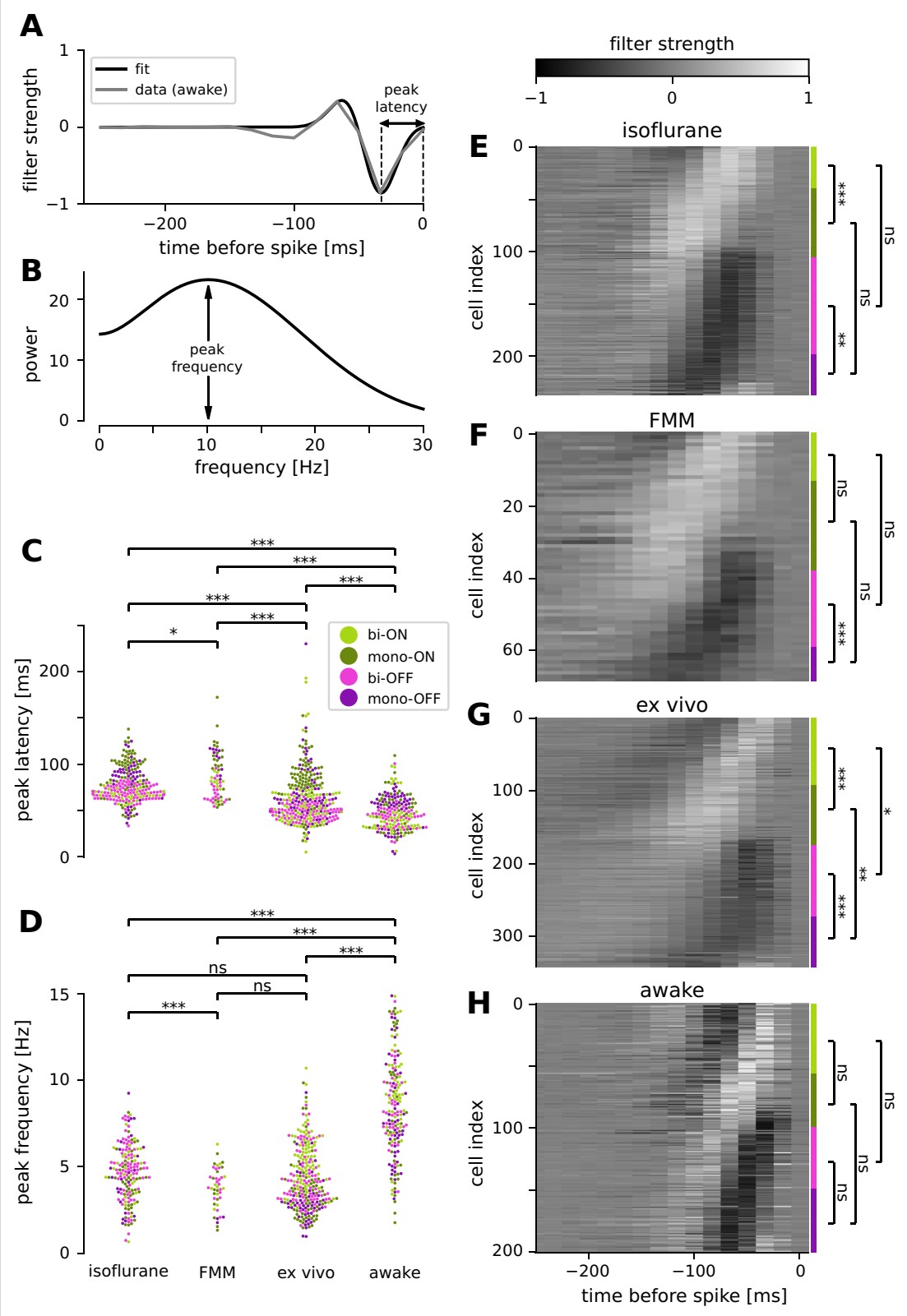

**Figure 5.** Retinal ganglion cells showed faster response dynamics in awake condition than in anesthetized or ex vivo conditions. (**A**) Temporal filter of a representative awake cell (gray, spike-triggered average [STA] of the full-field randomly flickering stimulus) and a difference-of-Gaussian curve fit (black) for estimating the latency of the first peak. (**B**) Power spectra of the example filter in A, based on the curve fit, for estimating the peak frequency. Population data of the peak latencies (**C**) and frequencies (**D**) across different conditions (light green, biphasic ON; dark green, monophasic ON; pink,

*Figure 5 continued on next page*

*Figure 5 continued*

biphasic OFF; violet, monophasic OFF). Here and thereafter, ***p < 0.001; **p < 0.01; *p < 0.05; ns, nonsignificant (*t*-test). The filter types were identified by the quadrants of the principal component analysis (PCA) biplot (see Methods for details). Population data of the temporal filters across different conditions: from top to bottom, isoflurane anesthesia (**E**, N = 238), fentanyl, medetomidine, and midazolam (FMM) anesthesia (**F**, N = 69), ex vivo (**G**, N = 342), and awake (**H**, N = 201). The four filter types are indicated on the right with corresponding colors.

curve (*Figure 5B*). Consistent with the results measured by the flickering stimuli at different frequencies (*Figure 4*), we found faster kinetics in awake animals than in anesthetized ones (*Figure 5C, D*): i.e., significantly shorter peak latencies (awake, 47 ± 18 ms, mean ± standard deviation; isoflurane, 77 ± 19 ms, p < 0.001, *t*-test; FMM, 87 ± 24 ms, p < 0.001; *Figure 5C*) and higher peak frequencies (awake, 8.5 ± 2.9 Hz; isoflurane, 4.6 ± 1.6 Hz, p < 0.001; FMM, 3.7 ± 1.2 Hz, p < 0.001; *Figure 5D*). Interestingly, we identified that the ex vivo data lied in-between. The ex vivo STAs had significantly longer peak latencies (64 ± 28 ms, p < 0.001) and lower peak frequencies (4.3 ± 1.8 Hz, p < 0.001) than the awake ones. In contrast, the peak latencies were significantly shorter in an isolated retinal preparation than in the anesthetized conditions (isoflurane, p < 0.001; FMM, p < 0.001), while the peak frequencies were comparable between these conditions (isoflurane, p = 0.3; FMM, p = 0.052).

We further analyzed the STA dynamics across different cell classes. For clustering the STAs obtained in each recording condition, we used the principal component analysis (PCA; see Methods for details). As in previous ex vivo studies (*Gollisch and Meister, 2008*; *Asari and Meister, 2014*), the first two principal components accounted for most of the variance (78–86%, collectively), and the four quadrants of the PCA biplot generally represented distinct filter shapes, corresponding to monophasic OFF (mono-OFF), biphasic OFF (bi-OFF), monophasic ON (mono-ON), and biphasic ON (bi-ON) response classes, respectively. In all recording conditions, no apparent cluster was found in this feature space, leading to a continuum of the STA shape patterns across populations (*Figure 5E–H*). Nevertheless, we identified two features that were different between ex vivo and in vivo, especially distinct in the awake condition. First, while the monophasic cells were generally slower than the biphasic ones, differences in the peak latencies were much larger in the ex vivo (*Figure 5G*; mono-ON 83 ± 24 ms versus bi-ON 58 ± 31 ms, p < 0.001, *t*-test; mono-OFF 67 ± 29 ms versus bi-OFF 48 ± 8 ms, p < 0.001) or anesthetized conditions (*Figure 5E*, isoflurane: mono-ON 91 ± 22 ms, bi-ON 67 ± 6 ms, p < 0.001; mono-OFF 89 ± 14 ms, bi-OFF 68 ± 8 ms, p < 0.001; *Figure 5F*, FMM: mono-ON 97 ± 30 ms, bi-ON 78 ± 11 ms, p = 0.051; mono-OFF 103 ± 13 ms, bi-OFF 70 ± 11 ms, p < 0.001) than in the awake condition (*Figure 5H*; mono-ON 51 ± 20 ms, bi-ON 44 ± 19 ms, p = 0.6; mon-OFF 51 ± 19 ms, bi-OFF 43 ± 12 ms, p = 0.1). Second, we found that ON cells were significantly slower than OFF cells ex vivo (mono-ON versus mono-OFF, p = 0.002; bi-ON versus bi-OFF, p = 0.03), but not in vivo (p > 0.2 in all the conditions examined). Taken together, our results suggest that awake responses are faster and their kinetics are more similar across different cell classes than ex vivo or anesthetized responses.

## Higher spiking activity in awake condition than ex vivo

How does a recording condition affect the firing properties of RGCs? The mean firing rate during the white-noise stimulus presentation was significantly higher in awake (34.9 ± 24.6 Hz; mean ± standard deviation) than in all the other recording conditions (isoflurane, 13.0 ± 13.1 Hz; FMM, 10.9 ± 10.7 Hz; ex vivo, 6.9 ± 6.6 Hz; all with p < 0.001, *t*-test in the logarithmic scale; *Figure 6A*). Importantly, this is presumably not due to differences in the stimulus condition. The maximum projected light intensity was comparable between in vivo (~31 mW/m$^2$ at the eye; see Methods) and ex vivo setups (~36 mW/m$^2$ on the isolated retina; *Vlasiuk and Asari, 2021*). Due to absorption by the eye optics, however, photon flux at the retina was estimated to be ~10 times lower for in vivo than for ex vivo recordings (on the order of $10^3$ and $10^4$ R*/photoreceptor/s, respectively; see Methods). Furthermore, awake mice show pupillary reflexes (*Figure 1—figure supplement 1*), leading to a smaller amount of light impinging on the retina than under anesthesia. Nevertheless, these differences in the lighting conditions cannot explain the higher firing and faster kinetics in the awake cells (*Figures 5 and 6A*) because RGCs typically show larger and faster responses to brighter and higher-contrast stimuli (*Baccus and Meister, 2002*; *Wang et al., 2011*).

For a further analysis, we examined the static nonlinear gain function of the recorded cells (*Equation 7* in Methods). This gain function accounts for nonlinearity associated with spike generation, such as spike threshold and firing rate saturation, and is generally well explained by a sigmoid function

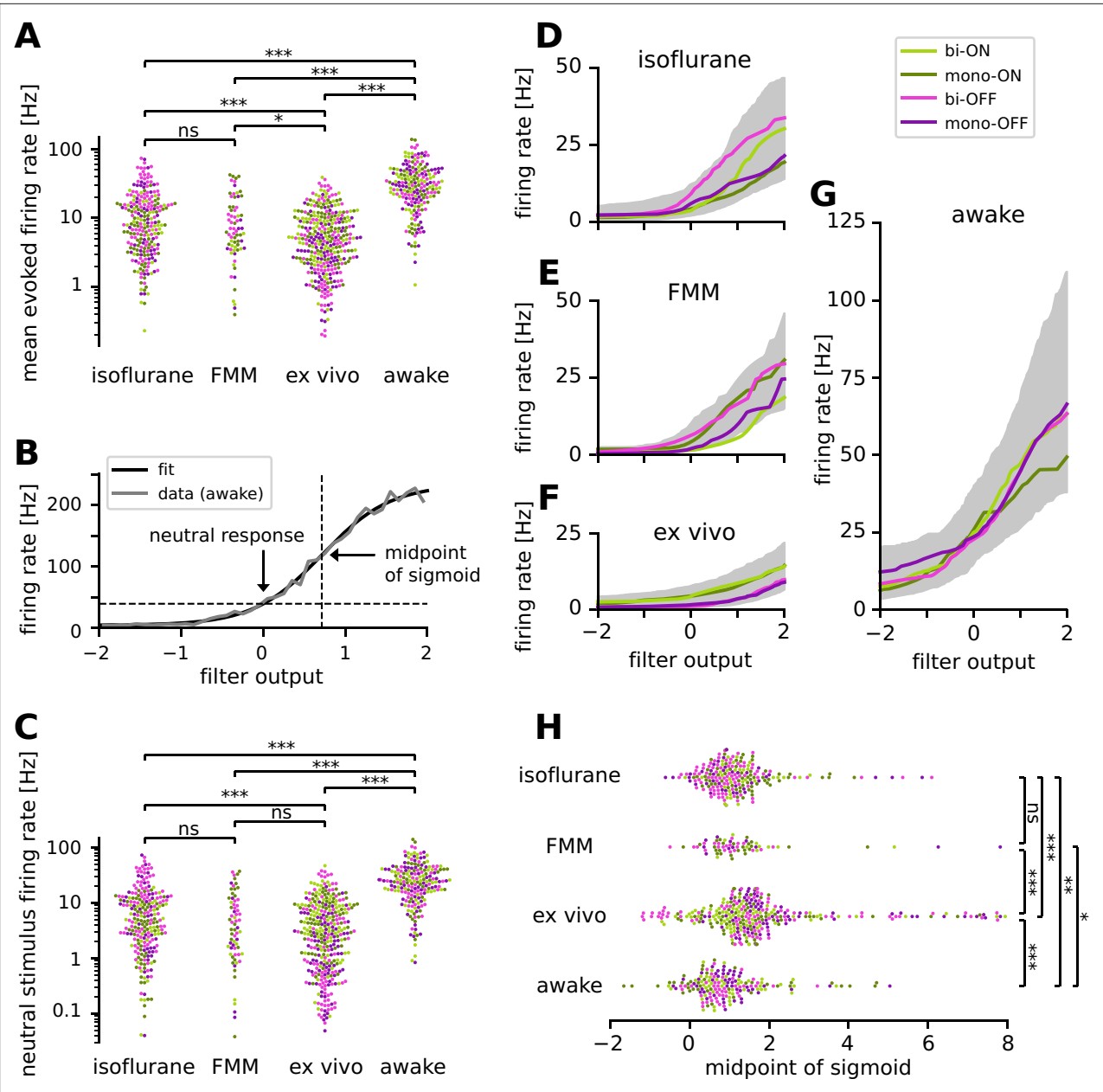

**Figure 6.** Retinal ganglion cells showed higher firing activity in awake condition than in anesthetized or ex vivo conditions. Cell classes are color coded as in *Figure 5* (light green, biphasic ON; dark green, monophasic ON; pink, biphasic OFF; violet, monophasic OFF). (**A**) Population data of the mean firing rates during the stimulus presentation period in four different recording conditions (isoflurane, fentanyl, medetomidine, and midazolam [FMM], ex vivo, and awake): ***p < 0.001; **p < 0.01; *p < 0.05; ns, nonsignificant (*t*-test on the logarithm of firing rates). (**B**) Static nonlinear gain function of a representative awake cell (the same one as in *Figure 5A*), estimated by the stimulus ensemble statistical techniques applied to the responses to a full-field randomly flickering stimulus (gray, *Equation 7* in Methods; black, sigmoid curve fit with the midpoint at 0.71). Note a high neutral stimulus response (40 Hz) defined as the firing rate at zero filter output (i.e., in the presence of stimuli orthogonal to the cell's spike-triggered average [STA]). (**C**) Population data of the neutral stimulus responses in each recording condition (in the same format as in A). Population data of the static nonlinear gain function (median for each cell type in corresponding colors; gray, interquartile range of all cells) across different conditions: isoflurane anesthesia (**D**, N = 238), FMM anesthesia (**E**, N = 69), ex vivo (**F**, N = 342), and awake condition (**G**, N = 201). (**H**) Population data of the midpoint of the sigmoid nonlinearity in each recording condition (Mann–Whitney *U*-test).

The online version of this article includes the following figure supplement(s) for figure 6:

**Figure supplement 1.** Batch effects in the data sets.

(see *Figure 6B* for example). For quantification, we assessed the neutral firing rate of the cells where the input to the gain function is zero, i.e., the firing rate in the presence of 'neutral' stimuli that are orthogonal to the cell's STA. Consistent with the results on the baseline firing rates measured by the sinusoidally flickering stimuli (*Figure 3*), the neutral firing rate was significantly higher in the awake condition (29.3 ± 22.5 Hz, mean ± standard deviation) than those under anesthesia (isoflurane, 9.5 ± 11.9 Hz, p < 0.001, *t*-test on a log scale; FMM, 7.5 ± 9.3 Hz, p < 0.001; *Figure 6C*). We also found that the neutral firing rates ex vivo (5.5 ± 7.1 Hz) were as low as those in the anesthetized conditions (isoflurane, p = 0.02; FMM, p = 0.1), and significantly lower than those in the awake condition (p < 0.001).

We further identified two distinct features in the gain function properties between ex vivo and in vivo responses. First, ON cells generally had a higher gain than OFF cells ex vivo (*Figure 6F*), whereas such cell-type-specific differences were not observed in vivo (isoflurane, *Figure 6D*; FMM, *Figure 6E*; awake, *Figure 6G*). Second, the midpoint of the sigmoid fitted to ex vivo responses (1.6 ± 1.4, median ± interquartile range) was significantly higher than that for in vivo responses (isoflurane, 1.1 ± 0.9; FMM, 1.1 ± 0.8; awake, 0.8 ± 1.0; all with p < 0.001, Mann–Whitney *U*-test; *Figure 6H*). In agreement with the outcomes obtained with the sinusoidally flickering stimulus (*Figure 3*), this suggests that in vivo responses are more linearized to cover a larger dynamic range, especially in the awake condition due to high neutral responses (*Figure 6G*). Indeed, while certain batch effects existed across recording sessions (*Figure 6—figure supplement 1*), the dynamic range computed from the gain function (see Methods) was generally largest in the awake condition (56 ± 54 Hz; median ± interquartile range) and significantly larger for in vivo (isoflurane, 26 ± 30 Hz; FMM, 29 ± 27 Hz) than for ex vivo (12 ± 12 Hz, all with p < 0.001, Mann–Whitney *U*-test).

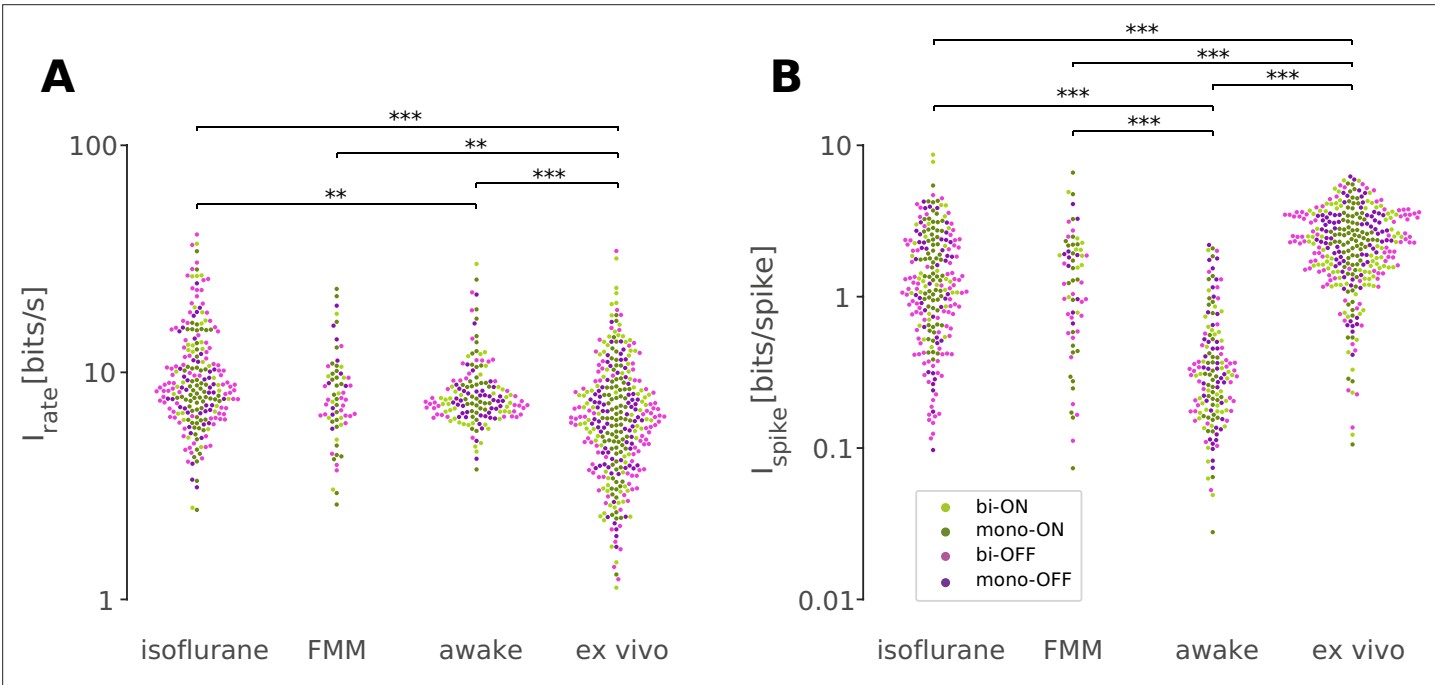

**Figure 7.** Retinal responses in vivo have a lower information rate in bits per spike but a higher rate in bits per second than those ex vivo. Information rate conveyed by retinal output spike trains in bits per second (**A**) and in bits per spike (**B**; *Equation 8* in Methods) in response to randomly flickering full-field stimuli under different recording conditions (from left to right: isoflurane, N = 231; FMM, N = 67; awake, N = 154; ex vivo, N = 328). Cell classes are color coded as in *Figures 5 and 6* (dark green, monophasic ON; light green, biphasic ON; pink, biphasic OFF; violet, monophasic OFF). ***p < 0.001; **p < 0.01 (Mann–Whitney *U*-test).

The online version of this article includes the following figure supplement(s) for figure 7:

**Figure supplement 1.** Rate coding was more preferable than latency coding for simulated awake retinal responses.

## Retinal outputs in awake mice are less efficient but more linearly decodable

Our experimental data demonstrated that retinal outputs in awake mice had distinct characteristics compared to those under anesthesia or ex vivo, such as higher baseline firing rates (*Figures 3 and 6*), faster response kinetics (*Figures 3–5*), and a larger dynamic range (*Figures 3 and 6* and *Figure 6— figure supplement 1*). What are the implications on retinal coding? To address this question, we quantitatively analyzed our data in three ways.

First, we took an information theoretic approach and calculated the information conveyed by the retinal output spike trains recorded under different conditions (*Figure 7*; see Methods). Overall, we found that in vivo responses had a slightly but significantly higher information rate in bits per second (*Figure 7A*; isoflurane, 8.6 ± 5.9 bits/s; FMM, 7.6 ± 3.6 bits/s; awake, 7.5 ± 2.2 bits/s; median ± inter-quartile range) than ex vivo responses (6.2 ± 1.9 bits/s; p < 0.001, p = 0.004, p < 0.001, respectively; Mann–Whitney *U*-test), but a lower information rate in bits per spike (*Figure 7B*; isoflurane, 1.2 ± 1.6 bits/spike; FMM, 1.3 ± 1.4 bits/spike; and awake, 0.3 ± 0.3 bits/spike; against ex vivo, 2.3 ± 1.9 bits/ spike; all with p < 0.001). In particular, the amount of information conveyed by a single spike in awake cells was the lowest due to high average firing rates (34 ± 25 Hz; mean ± standard deviation). This indicates that the retinal coding in vivo is less efficient than that ex vivo (*Schwartz, 2021*).

We next explored which neural coding scheme is favorable for awake retinal responses (*Figure 7— figure supplement 1*; see Methods). Specifically, we first simulated the retinal output responses to a step change in light intensity (e.g., *Figure 7—figure supplement 1A, B*) using a linear–nonlinear cascade model with the observed temporal filter (STA; *Figure 5*) and static nonlinear gain function (*Figure 6*). We then calculated the information based on the simulated peak firing rate ('rate code'; e.g., *Figure 7—figure supplement 1C*) or response latency ('temporal code'; e.g., *Figure 7—figure supplement 1D*). We found that the information estimated with the rate code was significantly larger for simulated awake responses (3.4 ± 1.2 bits; median ± interquartile range) than for anesthetized (isoflurane, 3.0 ± 0.8 bit, p <0 .001, Mann–Whitney *U*-test; FMM, 3.0 ± 0.8 bit, p = 0.011) or ex vivo responses (2.4 ± 1.3 bit, p < 0.001; *Figure 7—figure supplement 1E*). In contrast, information in the latency-based temporal code of the simulated awake responses (1.9 ± 1.1 bits) was comparable to that of ex vivo responses (1.7 ± 1.3 bits, p = 0.6), but significantly lower than those calculated from

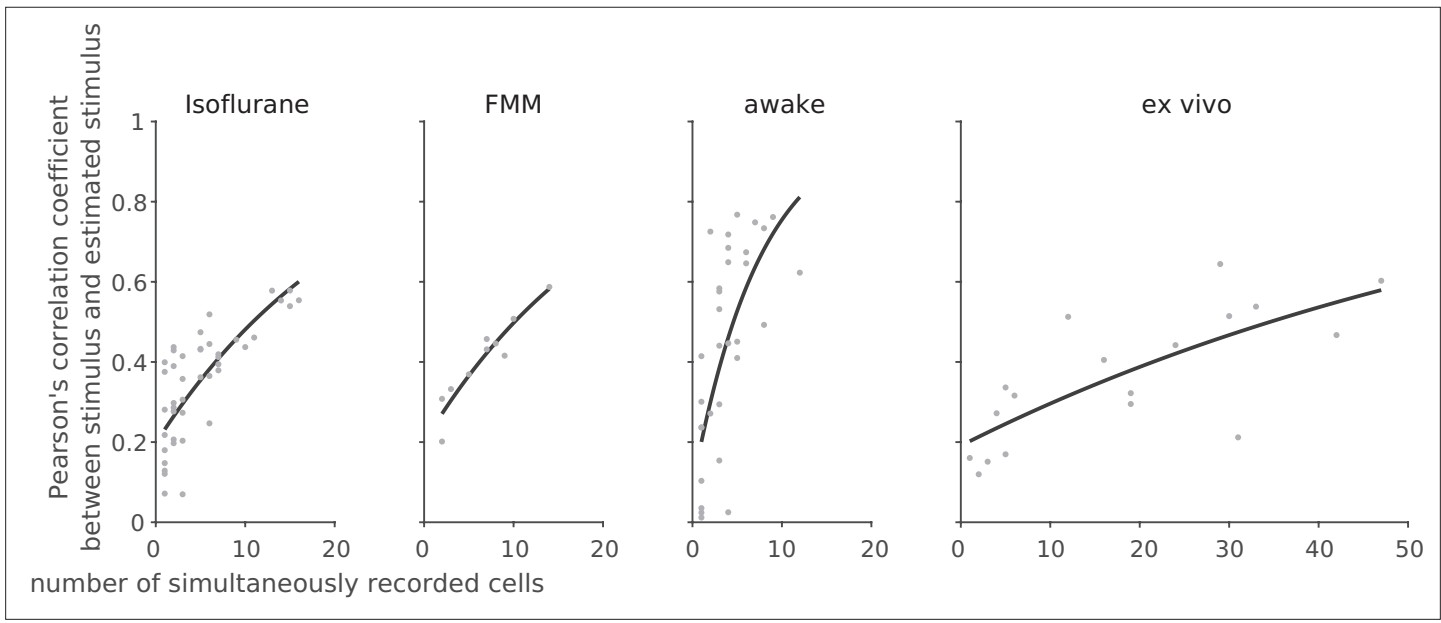

**Figure 8.** Linear decoding of retinal population responses worked best with awake responses. For each recording session, we performed linear decoding of the simultaneously recorded population activity (333 ms window) in response to the full-field randomly flickering stimulus (16.7 ms bin). As a measure of the decoding performance, we then calculated Pearson's correlation coefficient *R* between the presented and estimated stimuli with 10-fold cross-validation (from left to right: isoflurane, N = 44; FMM, N = 10; awake, N = 33; ex vivo, N = 18). The black line shows the exponential curve fit (*Equation 10* in Methods; isoflurane, a = 0.23 ± 0.04, b = 23 ± 7, estimate with 95% confidence interval; FMM, a = 0.24 ± 0.06, b = 21 ± 6; awake, a = 0.20 ± 0.13, b = 8 ± 4; ex vivo, a = 0.20 ± 0.10, b = 72 ± 36).

the simulated anesthetized responses (isoflurane, 2.3 ± 0.6 bits; FMM, 2.4 ± 0.6 bits; both with p < 0.001; *Figure 7—figure supplement 1F*). Here, we cannot directly compare the information between the rate and temporal codes as it depends on the assumptions made for the calculation (*Equation 9* in Methods). Nevertheless, information contained in a given coding scheme was calculated under the same assumptions across different recording conditions. Thus, our results suggest that the rate code is in favor of awake responses, while the temporal code is not.

Finally, to assess population coding properties, we performed a linear decoding analysis using simultaneously recorded population activity in response to the full-field randomly flickering stimulus (*Warland et al., 1997*; see Methods). For each recording session, the performance of the optimized linear decoder was evaluated by Pearson's correlation coefficient, $R$, between the presented and estimated stimuli with 10-fold cross-validation. For each recording condition, the decoding performance was then examined as a function of the number of simultaneously recorded cells by fitting an exponential saturation function (*Equation 10* in Methods; *Figure 8*). We found that single-cell linear decoding performance was comparable across recording conditions (awake, $R = 0.20 ± 0.13$, curve fit parameter $a$ with 95% confidence interval; isoflurane, 0.24 ± 0.04; FMM, 0.24 ± 0.06; ex vivo, 0.20 ± 0.10). At the population level, however, Pearson's $R$ reached 0.6 in about a third of the cases in the awake recordings with less than 10 cells to be considered, but none in the anesthetized recordings with less than 20 cells; and at least ~30 cells were required for the ex vivo recordings (*Figure 8*). Consistent with this observation, the exponential saturation constant was smaller for the awake responses ($b = 8 ± 4$ from *Equation 10*; an estimate with 95% confidence interval) than for the anesthetized (isoflurane, 23 ± 7; FMM, 21 ± 6) or ex vivo responses (72 ± 36). This indicates that retinal population responses in awake mice are more linearly decodable than in the other recording conditions, even though an image reconstruction is not necessarily the objective function of the retina.

## Discussion

Here, we established OT recordings in head-fixed mice (*Figure 1*) and systematically analyzed the retinal output properties in vivo using a standardized set of visual stimuli for characterizing retinal functions in a high-mesopic/low-photopic regime (*Baden et al., 2016*; *Jouty et al., 2018*). We found that awake response properties were overall similar to those under isoflurane or FMM anesthesia (*Figures 2–4*) or ex vivo (*Figures 5 and 6*); however, we identified three distinct features in the awake condition besides behavioral effects (*Figure 1—figure supplement 1*; *Figure 2—figure supplement 1*; *Schröder et al., 2020*): higher firing rates, a larger dynamic range (*Figures 3 and 6* and *Figure 6—figure supplement 1*) and faster response dynamics (*Figures 4 and 5*). These properties made the awake responses less efficient from an information theoretic viewpoint (*Figure 7* and *Figure 7—figure supplement 1*) but more linearly decodable at the population level (*Figure 8*). Our results highlight that our knowledge on retinal physiology cannot be simply translated from ex vivo to in vivo conditions, and that in vivo studies are indispensable to better understand retinal computation.

In awake animals, both the baseline and visually evoked firing rates of RGCs were much higher—on average by ~20 Hz—than in those under anesthesia or in an isolated preparation (*Figures 3 and 6*). To our knowledge, spiking activity of RGCs in awake mice has been reported only in some recent studies with a low sample size and none of them provided any descriptive statistics (*Hong et al., 2018*; *Schröder et al., 2020*; *Sibille et al., 2022*). Nevertheless, the reported examples (*Schröder et al., 2020*; *Sibille et al., 2022*) show high baseline activity (>10 Hz), consistent with our observation. These two studies employed high-density electrodes (NeuroPixels) to record axonal signals of RGCs in C57BL/6J mice, and are thus arguably comparable to our study from a methodological viewpoint. In contrast, using mesh electrodes directly injected into the mouse eye, *Hong et al., 2018* reported relatively low light-evoked RGC responses (<10 Hz) in albino CD1 strain. This is likely not due to the difference in the recording methods, but due to the mouse strain. Albino mice generally have poor vision with a reduced number of rod photoreceptors (*Abdeljalil et al., 2005*; *Brown and Wong, 2007*), suggesting a certain level of deficits in their retina. High firing activity in awake animals has been widely observed in many brain areas across species (*Greenberg et al., 2008*; *Sellers et al., 2015*; *Durand et al., 2016*; *Wright et al., 2017*; *De Franceschi and Solomon, 2018*; *Chen and Song, 2019*). Therefore, it could be a general feature of the brain in an awake state, including the retina.

This high firing activity had three consequences on the retinal output properties. First, by definition it leads to a lower DS/OS index value calculated as a normalized population vector (*Equation 3*

in Methods). Indeed, DS/OS cells were rather underrepresented in our awake recordings with high baseline activity (*Figures 2 and 3*), while overrepresented in our anesthetized recordings as well as in *Hong et al., 2018* with low baseline activity. Second, the baseline activity level affected the response kinetics (*Figures 4 and 5*). The higher the baseline firing is, the higher the conductance of the cell is, hence the shorter the response latency becomes (*Zohar et al., 2011*; *Wang et al., 2014*; *Durand et al., 2016*). Third, the dynamic range was also altered (*Figures 3 and 6* and *Figure 6—figure supplement 1*). Indeed, many awake ON cells exhibited an OFF-suppressive response, an emerging feature observed only in those cells with high baseline firing rates (~60 Hz). Here we categorized these cells separately from the other ON cells in our classification analysis (*Figure 2*). It is, however, unlikely that they form a distinct 'cell type' in the retina, even though we cannot deny a possibility that they were previously overlooked due to a difference in the recording modality (calcium imaging; *Baden et al., 2016*) or clustering methods (*Jouty et al., 2018*). Instead, these cells likely belong to a known cell type, and altered their response properties due to an increased baseline activity in the awake condition. Possible candidate cell types are: transient ON alpha cells (although they have a low baseline firing rate ex vivo; *Krieger et al., 2017*), ON contrast suppressive cells (although a sharp rebound response is missing in the calcium dynamics ex vivo; *Baden et al., 2016*), or suppressed-by-contrast cells (although their high baseline activity is suppressed by both ON and OFF stimuli; *Mastronarde, 1985*; *Tien et al., 2015*; *Jacoby and Schwartz, 2018*). It is interesting to identify the exact cell type of these RGCs and further characterize their function in vivo in future studies.

What are the mechanisms underlying such differences between in vivo and ex vivo retinal responses? One possibility is the difference in the physiological states of the retina. Even if ex vivo recordings are made under a proper temperature (*Vogel et al., 2016*; *Vlasiuk and Asari, 2021*) using a perfusion medium specifically formulated to support isolated retinal tissues (*Ames and Nesbett, 1981*), there are certain unavoidable differences to the physiological condition in vivo. In particular, the retinal pigment epithelium, a key player for the visual cycle and spatial buffering of ions, is often removed in an isolated retinal preparation. A lack of retinal supply can then alter the physiological states of the retinal neurons, potentially leading to a change in their visual response properties.

Differences in the input stimulus properties, in contrast, cannot explain our results, even though retinal responses depend a lot on stimulus conditions. For example, the higher the light intensity level is, the higher the temporal frequency sensitivity becomes for ex vivo RGC responses (threshold at around 20–30 Hz; *Wang et al., 2011*) and so does the critical flicker-fusion frequency at the behavioral level (15–40 Hz; *Umino et al., 2018*; *Nomura et al., 2019*). Response latencies depend also on the spatial patterns of stimuli as well as their contrast (*Bolz et al., 1982*; *Baccus and Meister, 2002*; *Sagdullaev and McCall, 2005*; *Pearson and Kerschensteiner, 2015*; *Tikidji-Hamburyan et al., 2015*). Here, we used full-field stimuli at a somewhat lower light intensity level for in vivo recordings than for ex vivo ones (see Methods for details). Pupillary reflexes in awake animals further decrease the dynamic range of the stimulus impinging on the retina (*Hattar et al., 2003*). Therefore, higher baseline activity and faster dynamics observed in awake animals cannot be simply attributed to the difference in the stimulus conditions.

Finally, when comparing retinal responses across different studies, one cannot ignore a possible effect of sampling bias because visual response dynamics are cell-type specific (*van Wyk et al., 2009*; *Krieger et al., 2017*; *Ravi et al., 2018*; *Tengölics et al., 2019*). In an extreme case, delayed ON cells have the response latency slower than other RGC types by hundreds of milliseconds in an isolated mouse retina (*Mani and Schwartz, 2017*). Response dynamics are also species specific. For example, ON RGCs show higher baseline activity and faster responses than OFF cells in the macaque (*Chichilnisky and Kalmar, 2002*) and guinea pig (*Zaghloul et al., 2003*) retinas, whereas slower in the salamander retina (*Gollisch and Meister, 2008*), and such asymmetry between ON and OFF dynamics is pathway specific in the rat retina (*Ravi et al., 2018*). Moreover, kinetics can vary even for the same cell type, depending on the retinal location (*Warwick et al., 2018*) or lilighght-adaptatiot-adaptation state (*Chang and He, 2014*; *Tengölics et al., 2019*). Sampling bias can then skew the results in many different ways. Given that most RGCs project their axons through the OT (*Ellis et al., 2016*; *Román Rosón et al., 2019*), we expect that the OT recordings employed in this study should have access to most cell types, if not all. Moreover, here we observed a wide range of visual responses (*Figure 2*), and no apparent difference was found in the STA distributions across different recording conditions

(*Figure 5*). We thus expect that the effects of sampling bias should be minimal on the observed differences between in vivo and ex vivo retinal response characteristics.

Taken together, our findings indicate that clear differences exist in retinal physiology between in vivo and ex vivo. This highlights the importance of studying retinal function in vivo, especially in the fully physiological awake condition.

## Implications on retinal coding in vivo

The optic nerve fiber forms an information bottleneck in the early visual system. The human retina, for example, contains ~$10^8$ input neurons (photoreceptors) but only ~$10^6$ output neurons (RGCs) whose axons make up the optic nerve (*Masland, 2012*; *Sanes and Masland, 2015*). The retina has then been assumed to optimize the channel capacity by compressing the visual information as much as possible and conveying signals to the brain using as few spikes as possible (*Attneave, 1954*; *Barlow and Rosenblith, 1961*; *Laughlin, 2001*). This so-called efficient coding hypothesis is supported by both ex vivo experimental data and theoretical analyses. Under ex vivo conditions, while some RGCs exhibit high firing under certain conditions (*Ke et al., 2014*; *Smeds et al., 2019*), many RGCs are silent most of the time and fire spikes at high rates only when their selective stimulus features are presented (*Gollisch and Meister, 2010*; *Baden et al., 2016*; *Jouty et al., 2018*). This sparse RGC response, ensured by low spontaneous activity and strong nonlinearity (*Pitkow and Meister, 2012*), helps achieve high efficiency and low redundancy in the visual representation of the retina (*Doi et al., 2012*). Furthermore, efficient coding principles can explain various physiological properties of the retina, such as separation of retinal outputs into multiple cell types (*Gjorgjieva et al., 2019*), even though it cannot fully account for the retinal function (*Schwartz, 2021*).

The retinal output characteristics we observed in vivo, however, provide a different view on the retinal code. First, unlike ex vivo conditions, RGCs in awake mice generally showed a high baseline activity with more linearized responses (*Figures 3 and 6*). Consequently, many awake ON cells primarily encoded light decrements by suppression from high baseline, rather than light increments by activation from low baseline. These cells convey the same information as conventional ON cells do—the more spikes, the higher light intensity—but using much more spikes. Thus, the amount of information transmitted per spike is much lower in vivo than ex vivo (*Figure 7*), violating the efficient coding principles. Second, we found less variability in the visual response dynamics across different RGC classes in vivo (*Figure 5E–H*). This suggests that temporal coding framework may not be readily applicable in vivo as it requires noticeable differences in spike timing patterns or response latencies between two or more channels, such as ON and OFF pathways converging onto ON–OFF RGCs (*Gollisch and Meister, 2008*). Our modeling analysis indeed demonstrated that awake responses were in favor of a rate code but not a latency-based temporal code (*Figure 7—figure supplement 1*), and that a linear decoder worked significantly better with the population responses in awake mice (*Figure 8*), even though an image reconstruction is not necessarily the retina's objective function. These results point out a need to reconsider retinal function in vivo.

What are the advantages of such energy-inefficient retinal coding? As shown in our awake recordings, one can gain faster response kinetics (*Figure 5*) and wider bandwidths (*Figure 6*). In addition, redundancy in the dense code helps transmit signals accurately even with intrinsically noisy spike trains, as demonstrated by a high performance of a simple linear decoder (*Figure 8*). These features are all highly beneficial from a behavioral viewpoint—for example, to detect predators robustly and promptly—and thus worth achieving for survival at an expense of energy cost. Interestingly, dense coding can be a general feature of early sensory processing, as suggested by high baseline firing in vestibular and cochlear nuclei (*Fuchs and Kimm, 1975*; *Rhode and Smith, 1986*; *Warchol and Dallos, 1990*; *Beraneck and Cullen, 2007*), tonic activity in hair cells (*Sachs and Abbas, 1974*; *Wu et al., 2016*), and dark current in vertebrate photoreceptors (*Hagins et al., 1970*; *Okawa et al., 2008*). In fact, the energy cost of dense retinal coding may not be so problematic with a relatively small number of neurons, as opposed to the total energy required for operating ~$10^9$ neurons in the cortex where sparse coding has certain advantages from the viewpoints of both energy efficiency and sensory processing (*Olshausen and Field, 1996*; *Asari et al., 2006*).

The retina, however, may employ different coding principles under different light levels (*Wang et al., 2011*; *Tikidji-Hamburyan et al., 2015*; *Borghuis et al., 2018*). All our in vivo recordings were done in a high-mesopic/low-photopic range (at ~$10^3$ R*/photoreceptor/s; see Methods). In a scotopic

condition, in contrast, ex vivo studies have reported a high tonic firing rate for OFF RGC types (~50 Hz, OFF sustained alpha cells in mice; *Smeds et al., 2019*; ~20 Hz, OFF parasol cells in primates; *Ala-Laurila and Rieke, 2014*), while a low tonic firing for corresponding ON cell types (<1 Hz). Since these ON cells were shown to be responsible for light detection by firing in the silent background (*Smeds et al., 2019*), here the retina seems to operate with a sparse feature detection strategy. Further characterizations of the retinal code in vivo and its behavioral relevance will be needed to clarify what exactly the eye tells the brain.

### Effects of anesthetics in early visual system

Anesthetics generally lower the excitability of nerve cells: for example, isoflurane acts on the gamma-aminobutyric acid (GABA) type A receptors to silence the brain (*Jenkins et al., 1999*). It thus makes sense that the retinal outputs were reduced under anesthesia (*Figures 3 and 6*) and showed slower dynamics (*Figures 4 and 5*) than in awake animals. It is, however, difficult to generalize the effects of anesthesia because multiple mechanisms of action are involved, exerting complex effects on the sensory systems in anesthetic- and dose-dependent manners (*Populin, 2005*). For example, while visual cortical responses are reduced under anesthesia (*Haider et al., 2013*; *Vaiceliunaite et al., 2013*), auditory responses are enhanced (*Raz et al., 2014*; *Sellers et al., 2015*) and the response latency becomes shorter in the auditory pathway (*Ter-Mikaelian et al., 2007*).

Our results are nevertheless consistent with previous studies on the effects of anesthesia in the early visual system. Specifically, anesthesia reduces the overall activity of both SC and dLGN neurons, and leads to multiple changes in temporal processing of these two major retinorecipient areas, such as reduced sensitivity to high temporal frequencies and longer response latencies (*Zhao et al., 2014*; *Durand et al., 2016*; *De Franceschi and Solomon, 2018*). While different anesthetics were used in these studies (urethane as opposed to isoflurane and FMM), our consistent findings suggest that such changes in the response dynamics arise largely from the retina. In contrast, the effects of anesthesia on spatial response properties seem more complex. While spatial processing remains largely intact in dLGN (*Durand et al., 2016*), lower sensitivity to contrast and larger receptive field sizes were reported in SC under urethane anesthesia (*De Franceschi and Solomon, 2018*). Moreover, SC neurons show increased orientation selectivity (OS) under isoflurane anesthesia (*Kasai and Isa, 2022*). Here we also found a larger fraction of DS/OS cells under both isoflurane and FMM anesthesia than in the awake condition (*Figure 2D–F*). This, however, could be due to the difference in the baseline firing rate (*Figure 3*) as the DS/OS index is defined by a normalized population vector in response to moving gratings (*Equation 3* in Methods; *Mazurek et al., 2014*). It is a future challenge for fully characterizing spatiotemporal processing of the retina in vivo.

# Materials and methods

No statistical method was used to predetermine the sample size. The significance level was 0.05 (with Bonferroni correction where appropriate) in all analyses unless noted otherwise. All experiments were performed under the license 233/2017-PR from the Italian Ministry of Health, following protocols approved by the Institutional Animal Care and Use Committee at European Molecular Biology Laboratory. The data analyses were done in Python.

### Animals

A total of 55 female C57BL/6J mice were used (chronic, 3; acute, 35; failures, 17, including 10 initial unsuccessful attempts to record from the optic chiasm) at around 3 months of age at the time of the surgery. Mice were kept on a 12-hr light/12-hr dark cycle and given water and food ad libitum. After the chronic implantation of electrodes, the animals were kept single-housed.

### Chronic recordings

For implanting electrodes, animals were anesthetized (induction, 4% isoflurane; maintenance, 1–2%) and placed inside a stereotaxic apparatus (Neurostar). Throughout the surgery, temperature was maintained at 37°C using a heating pad (Supertech Physiological), and eye ointment (VitA-POS) was used to prevent the eyes from drying. After positioning the mouse head, the scalp skin was disinfected (Betadine) and removed with scissors. Soft tissue was removed from the skull surface with

a round scalpel, and ethanol and acetone were applied to the skull for disinfection and removal of any residual compounds. The skull was glazed with a drop of cyanoacrylate adhesive (Loctite 401) to secure in place the surrounding skin, and then registered in the stereotaxic controller software (NeuroStar). A dental drill (WPI) was used to leave three marks on the skull to label the entry point for targeting the OT: [−1.34,+1.87,+4.74], [−1.70,+1.87,+4.74], and [−1.82,+2.35,+4.07] in [Anterior–Posterior, Medial–Lateral, Dorsal–Ventral] coordinates, respectively. A hole was drilled for a reference silver wire (A-M Systems) above the cerebellum, and the wire was inserted sideways to avoid excessive brain damage. After covering the hole and wire with Vaseline, the wire was attached to the skull with cyanoacrylate adhesive and dental cement (Paladur, PALA). Subsequently, a custom-designed titanium head-plate was attached to the skull with dental cement, followed by a craniotomy (diameter, 1–2 mm) and durotomy. A chronic silicone probe (Buzsaki32L, NeuroNexus) was then inserted with the stereotaxic controller (75 μm/min) first to a depth of 2 mm, retracted by 1 mm to release pressure created by the initial brain entry, and then to a depth of 4.5 mm from the skull surface. After reaching the desired depth, a microdrive (dDrive, NeuroNexus) was attached to the skull with dental cement, and the probe was removed from its mount and covered with a protective cap. The cables and connectors were cemented to the cap and covered with paraffin film (Bemis, Parafilm) to prevent the mouse from damaging the implant.

After the surgery, the animals were kept on a heating pad (Sera, Thermo comfort mat S) until they recovered from anesthesia. During the next 4 days, the mice received an anti-inflammatory/antibiotic cocktail (Rimadyl/Baytril; 0.5 mg/ml each, 0.01 ml/g). The antibiotic (Baytril) was given for an additional 3 days through drinking water (0.17 mg/ml).

After a recovery period of 5 days, the mice were placed on a custom-made rotary treadmill (diameter, 20 cm) with their head fixed for the recordings (at most two times a day, each for <2 hr). During the initial sessions, we moved the probe until visual responses were observed (*Figure 1A–D*). The electrophysiology data were recorded at 30 kHz from each recording site (SmartBox, NeuroNexus) together with synchronization pulses from the visual stimulation device (see below). In total, we made 17 recording sessions from 3 out of 6 animals, where up to 10 cells were simultaneously recorded in each session (4.4 ± 2.4 cells/session, mean ± standard deviation).

After all the recording sessions, the electrode position was verified histologically (e.g., *Figure 1E*). The mice were anesthetized (2.5% Avertin, 16 μl/g, intraperitoneal injection) and perfused with paraformaldehyde (PFA; 4% in phosphate buffer solution). The animal's head with the electrode left in position was collected without the skull base, and stored in fixative solution (4% PFA) at 4°C for at least four days. This helped the brain tissue harden around the silicon probe, hence leaving a visible mark after removing the probe. Harvested brain tissue was then coronally sliced with a vibratome (Leica, VT1000S; thickness, 150 μm) and imaged under a bright-field microscope (Leica, LMD7000).

## Acute recordings

Animals were anesthetized and placed inside a stereotaxic apparatus with a heating pad as described above. A contact lens (diameter, 3 mm; Ocuscience) was used for the target eye to prevent it from drying. The scalp was removed, registered in the stereotaxic controller, and the three entry points for targeting the OT were marked with the stereotaxic drill. A well was made with dental cement around the marks later to hold saline solution on top of the brain. After a craniotomy (diameter, 2 mm), an acute silicone probe (Buzsaki32L, Neuronexus) coated with a fluorescent dye (DiI stain, Invitrogen, D282) was slowly inserted into the brain (100 μm/min) with a micromanipulator (Sensapex, SMX) attached to the stereotaxic apparatus. While approaching the target depth, visual stimuli were presented (full-field contrast-inverting stimulus at 0.5 Hz; see below). The probe was moved until a maximum number of visually responsive cells were seen at once (up to 20 cells; 7.7 ± 5.3 cells/session, mean ± standard deviation; 52 sessions in total from 23 out of 27 animals).

Throughout the recordings, the mouse was kept under anesthesia with 1% isoflurane. The depth of anesthesia was monitored by the breathing rate (~1 breath/s). Alternatively, we used a cocktail of FMM during the recordings. In this case, once the electrode was in position, an FMM solution (fentanyl, 0.05 mg/kg; medetomidine, 0.5 mg/kg; midazolam, 5 mg/kg; in 0.9% NaCl saline) was intraperitoneally administered, and the isoflurane dose was progressively decreased to 0%. Buprenorphine (0.1 mg/kg) was injected 20 min after the termination of isoflurane, and the recording session was initiated 10 min after the buprenorphine injection. The depth of anesthesia was monitored through

the heart rate (below around 300 beats per minute) and supplemental FMM anesthesia was provided when required. The electrophysiology data and the visual stimulation signals were recorded at 30 kHz/electrode (SmartBox, NeuroNexus).

At the end of the recording session, the electrode location was verified histologically. After retracting the silicone probe, the animal was perfused as described above. The brain tissue was harvested and post-fixed overnight in 4% PFA at 4°C. Coronal sections of the brain tissue (thickness, 150 μm) were then examined under a fluorescence microscope (Leica, LMD7000 with N2.1 filter cube) to visualize the trace left by the DiI stain on the probe.

For acute awake recordings, we first performed head-plate implantation as described above. In short, after placing an animal on a stereotaxic apparatus under anesthesia, we (1) surgically removed the skin and the periosteum above the skull; (2) attached a head-plate to the skull with dental cement; and (3) applied a thin layer of cyanoacrylate adhesive (Loctite 401) and a thick layer of silicone sealant (Kwik-cast) to provide mechanical and thermal support. Animals were then allowed to recover for a week, and subsequently habituated over the course of 10 days to head fixation in the recording setup as described above. On the day of the recording, we removed the silicone to expose the skull surface of the animal with its head fixed in the recording setup, and determined the electrode penetration path to the target area using the robotic stereotaxic system (StereoDrive, NeuroStar). The animal was then briefly anesthetized (for about 5 min) and a hole was drilled around the electrode entry point on the skull. After the removal of the anesthesia, the electrode attached to the robotic arm was lowered at 5 μm/s until visual responses were found in the target area. Recordings and histological verification of the electrode location were performed as described above.

## Visual stimulation

Visual stimuli were presented by a custom gamma-corrected digital light processing (DLP) device (Texas Instruments, DLPDLCR3010EVM-LC) where the original green and red light-emitting diodes (LEDs) were replaced with ultra-violet (UV; 365 nm; LZ1-00UV00, LED Engine) and infrared (IR; 950 nm; SFH 4725S, Osram) LEDs, respectively. The UV and blue (465 nm) lights were projected onto a spherical screen (radius, 20 cm) with UV-reflective white paint (waterfowl store) placed ~20 cm to the contralateral side of an animal's eye from the implanted probe (*Figure 1A*), whereas the IR light was used as synchronization pulses, recorded via a photodiode with a custom transimpedance amplifier. The visual stimuli (1280-by-720 pixels; frame rate, 60 Hz) covered $\omega_{azimuth}$ = 73° in azimuth and $\omega_{azimuth}$ = 44° in altitude from the mouse eye position (*Figure 1B*). The maximum light intensity at the eye position was 31.3 mW/m² ($E_{365}$ = 15.4 mW/m² for UV LED and $E_{365}$ = 15.9 mW/m² for blue LED; measured with S121C sensor, Thorlabs), leading to mesopic to photopic conditions (see also Light intensity).

Using this DLP setup and QDSpy software (*Franke et al., 2019*), we presented the following set of visual stimuli: a randomly flickering full-field stimulus (5 min; 60 Hz), a moving grating stimulus (spatial frequencies of square waves, 3° or 20°; moving speed, 7.5°/s or 15°/s, in eight different directions), a full-field stimulus whose intensity followed a sinusoid (1.5 Hz) with a linearly increasing amplitude from 0% to 100% contrast over 10 s (10 trials; *Figure 3*), and a full-field sinusoidally flickering stimulus (maximum contrast) at different temporal frequencies (1.875, 3.75, and 7.5 Hz, each for 2 s; 15 Hz for 1 s; 10 trials; *Figure 4*). The last two full-field stimuli were preceded by a sequence of 'OFF–ON–OFF' stimulation at maximum contrast (full-field contrast inversion; 2 s each) and interleaved by a 1-s-long gray screen across trials. These stimuli were equivalent to those used for differentiating ~30 RGC types ex vivo (*Baden et al., 2016*; *Jouty et al., 2018*).

## Light intensity

We calculated the photon flux on the mouse retina in vivo as: $I_\lambda = T_\lambda \cdot F_\lambda \cdot A_{pupil}/A_{retina}$, where $T_\lambda$ is the transmittance of the eye optics at wavelength $\lambda$; $F_\lambda = E_\lambda/(hc/\lambda)$ is the photon flux at the pupil with Planck's constant $h$ and the speed of light $c$; $A_{pupil} \approx \pi \cdot (d_{pupil}/2)^2$ is the pupil area with diameter $d_{pupil}$; and $A_{retina} \approx (\pi \cdot d_{retina} \cdot \omega_{azimuth}/360°) \cdot (\pi \cdot d_{retina} \cdot \omega_{azimuth}/360°)$ is the illuminated retinal area with diameter $d_{retina}$. Assuming that $T_{365}$ = 50%, $T_{454}$ = 68% (*Jacobs and Williams, 2007*), $d_{pupil}$ = 1 mm and $d_{retina}$ = 4 mm, we then obtained $I_{365}$ = 3.81 × 10³ and $I_{454}$ = 6.64 × 10³ photons/s/μm². Given a total photon collecting area $a$ = 0.2 and 0.5 μm² for cones and rods, respectively (*Nikonov et al., 2006*), and a relative sensitivity $s_{365}$ = 90% and $s_{454}$=0% for S-cones while $s_{365}$ = 25% and $s_{454}$ = 60% for M-cones and rods (*Jacobs and*

*Williams, 2007*), we estimated the photoisomerization (R*) rate as: $a \cdot \sum_\lambda I_\lambda s_\lambda = 2.5 \times 10^3$ R*/rod/s, $0.7 \times 10^3$ R*/S-cone/s, and $1.0 \times 10^3$ R*/M-cone/s.

In contrast, the maximum light intensity of the ex vivo setup was $E = 36$ mW/m² (*Vlasiuk and Asari, 2021*). Assuming that the white light from a cathode-ray tube monitor is centered around $\lambda = 500$ nm, we estimated the photon flux on an isolated retina as: $I = E/(hc/\lambda) = 9 \times 10^4$ photons/s/µm². Given the sensitivity $s_{500} = 2\%$ for S-cones and 40% for M-cones and rods (*Nikonov et al., 2006*), we then obtained: $a \cdot I \cdot s_{500} = 4 \times 10^4$ R*/rod/s, $2 \times 10^3$ R*/S-cone/s, and $4 \times 10^4$ R*/S-cone/s. Thus, the light intensity level was about an order of magnitude higher for ex vivo data than for in vivo data.

## Physiology data analysis

In total, we obtained 282 cells in awake conditions (75, chronic; 282, acute), and 325 and 103 cells in acute anesthetized recordings with isoflurane and FMM, respectively. Spike sorting was performed with SpykingCircus (*Yger et al., 2018*) for semi-automatic cluster detection and Phy (*Rossant, 2020*) for data curation. In brief, spikes were detected from highpass-filtered data traces (cutoff, 300 Hz) as those events 7 standard deviations away from the mean. Single units were then identified by clustering in principal component space, followed by manual inspection of spike shape as well as auto- and cross-correlograms. As expected from axonal signals (*Barry, 2015*), spike waveforms typically had a triphasic shape (e.g., *Figure 1F–H*). No putative single units had the interspike intervals below 1ms for a refractory period (e.g., *Figure 1I–K*), except for 1 (out of 103) for FMM-anesthetized recordings. Most units had a minimum interspike interval above 2ms (93%, awake; 94%, isoflurane; 99%, FMM). Here, we included in the analysis those small numbers of visually responsive cells that had a small fraction of interspike intervals below 2 ms; removing them did not affect the conclusions. Not all cells responded to the entire stimulus set, but cells responding to any of the presented stimuli were included in the analysis.

For ex vivo recordings, we reanalyzed the data sets in *Vlasiuk and Asari, 2021*. Specifically, the ex vivo data sets included the activity of 696 cells recorded with a multi-electrode array (from 18 isolated mouse retinas) in oxygenated Ames' medium at 37°C in response to a randomly flickering full-field visual stimulus projected from a gamma-corrected cathode-ray tube monitor (frame rate, 100 Hz) or a DLP device (60 Hz). Given that the temperature of the ocular surface is ~37°C (*Vogel et al., 2016*), the retina likely operates at around 37C in vivo.

## Response quality

We assessed the cell's response quality based on the trial-to-trial reliability of the response $r(t)$ during the 'ON–OFF' period of the OFF–ON–OFF stimulus sequence (see *Figure 2A* for example). Specifically, the signal-to-noise ratio was calculated for each cell as follows:

$$\text{SNR} = \frac{\text{var}\left[\langle r(t) \rangle\right]_t}{\left\langle \text{var}\left[r(t)\right]_t \right\rangle}, \tag{1}$$

where $\langle \cdot \rangle$ indicates the mean over trials, and $\text{var}[\cdot]_t$ the variance over time $t$ (bin size, $\Delta t = 16.6$ ms). We set a threshold at 0.15 to select reliably responsive cells for further analyses (*Figure 2B*).

## Response polarity

To characterize the cell's preference to stimulus polarity, we defined an ON–OFF index using the responses to the full-field contrast inversion:

$$\text{ON-OFF index} = \frac{r_{\text{ON}} - r_{\text{OFF}}}{r_{\text{ON}} + r_{\text{OFF}}}, \tag{2}$$

where $r_{\text{ON}}$ and $r_{\text{OFF}}$ are the mean firing rate during the ON and the second OFF periods of the OFF–ON–OFF stimulus sequence, respectively. Positive and negative ON–OFF index values indicate stronger responses to ON and OFF stimuli, respectively (*Figure 2C*).

## DS and OS

DS and OS indices were calculated by projecting the responses to the moving grating stimuli onto a complex exponential (*Mazurek et al., 2014*):

$$\text{DS or OS index} = \left\| \frac{\sum_k e^{-i\alpha\omega_k} \cdot r_k}{\sum_k r_k} \right\|, \tag{3}$$

where $\omega_k$ and $r_k$ are the angle of the $k$th direction and the cell's corresponding responses, respectively; and $\alpha = 1$ and 2 for the DS and OS indices, respectively. Cells were considered DS and/or OS when the corresponding index was higher than 0.15 and p < 0.2 calculated by bootstrap methods (1000 repetitions; *Figure 2D–F*).

## Temporal frequency sensitivity

Temporal frequency sensitivity was assessed by fitting an even power of sine function to the responses to the full-field sinusoidally flickering stimulus at different frequencies $h$ (1.875, 3.75, 7.5, or 15 Hz; *Figure 4*):

$$s(t) = A \cdot \sin^q \left( \pi h t - \frac{\phi}{2} \right) + B, \tag{4}$$

where $A$ and $B \geq 0$ are the evoked response amplitude and baseline activity, respectively; and $q = 2n$ with $n \in N^+$ is the exponent of the sine function. The phase $\phi \in [-\pi, \pi)$ indicates the relative position between the response peak and the sinusoidal stimulus patterns. For example, $\phi = -\pi/2$ and $\pi/2$ are obtained if the response reaches its peak when the stimulus is brightest and darkest, respectively.

The fit quality was then assessed by the explained variance:

$$R^2 = 1 - \frac{\text{var}[\text{Data} - \text{Fit}]}{\text{var}[\text{Data}]}. \tag{5}$$

We set a threshold at 0.2 to select cells with a good fit hence responsive to the stimulus (*Figure 4C*), and compared the proportions of the responsive cells across different recording conditions (two-proportion z-test; *Figure 4D*).

## Sensitivity to contrast

We used an even power of sine function with a sigmoid envelope to characterize the responses to the flickering stimulus ($h$ = 1.5 Hz) with increasing contrast (*Figure 3*):

$$S(t) = \frac{A}{1 + e^{-\lambda(t - t_0)}} \cdot \sin^q \left( \pi h t - \frac{\phi}{2} \right) + B, \tag{6}$$

where $t_0$ and $\lambda > 0$ are the midpoint and the steepness of the sigmoid, respectively; and the other free parameters are the same as $s(t)$ in *Equation 4*. The fitted parameter values were then compared across different recording conditions: Mann–Whitney $U$-test for the baseline $B$ (*Figure 3D*) and the absolute amplitude $|A|$ (*Figure 3E*), and t-test for the phase $\phi$ for ON and OFF responses, respectively ($k$-means clustering with $k$ = 2; *Figure 3F*). As a measure of the sensitivity to contrast, we used an estimated response magnitude at 10% contrast from *Equation 6* as well as the absolute slope size at the midpoint of sigmoid ($|A\lambda/4|$).

## Temporal filters and static nonlinear gain functions

For systematically characterizing the response dynamics, we used stimulus ensemble statistical techniques ('reverse correlation' methods; 500 ms window) to calculate the linear filter (*Figure 5*) and static nonlinear gain function (*Figure 6*) of the recorded cells in response to a randomly flickering visual stimulus (*Meister et al., 1994*; *Chichilnisky, 2001*). First, we obtained the linear filter of each cell by calculating an STA of the stimulus with ±1 being 'white' and 'black', respectively. As a quality measure, p-value was computed for each time bin against a null hypothesis that the STA follows a normal distribution with a mean of zero and a variance of $1/C$, where $C$ is the total number of spikes. As a measure of the cell's temporal frequency tuning, we then estimated the peak latency by fitting a difference-of-Gaussian curve to the linear filter (e.g., *Figure 5A*); and the spectral peak frequency by the Fourier analysis on the fitted curve (e.g., *Figure 5B*). The curve fitting quality was assessed by the explained variance $R^2$ as in *Equation 5*. We discarded the cells if p > $10^{-18}$ for all time bins or $R^2$ < 0.8. We ran a t-test to compare the temporal frequency tuning properties at the population level across recording

conditions (*Figure 5C, D*) or across different cell types in each recording condition (*Figure 5E–H*). Spatial response properties were not examined in this study because complex nonlinear kinematics of eye movements in the awake condition precluded the analysis at a high enough spatial resolution; and because it has been shown that anesthesia has no effect on the spatial processing in dLGN, a direct downstream of the retina (*Durand et al., 2016*).

Static nonlinear gain function *P*(response|stimulus) of each cell was computed by taking the ratio between the distribution of spike-triggered stimulus ensembles *N*(stimulus|response) projected onto the *L2*-normalized STA (bin size, 0.1) and that of the entire stimulus ensembles *N*(stimulus) (e.g., *Figure 6B*):

$$P\left(\text{response} \mid \text{stimulus}\right) = \frac{N\left(\text{stimulus} \mid \text{response}\right)}{N\left(\text{stimulus}\right)} \Big/ \Delta t. \tag{7}$$

A sigmoid function was fitted to P(response|stimulus) for smoothing. The neutral stimulus response was then defined as the vertical-intercept of the sigmoid function at the horizontal axis value of zero: that is, the firing rate when the stimulus is orthogonal to the STA. We ran a *t*-test to compare the neutral stimulus responses and the average firing rates during the stimulus presentation at the population level across conditions (*Figure 6A, C*); and Mann–Whitney *U*-test to compare the midpoint of the sigmoid (*Figure 6H*). The dynamic range was computed for each cell as the difference between the maximum and minimum firing rates of the static nonlinear gain function (*Figure 6—figure supplement 1*).

## Response classification

In this study, we were not able to perform a morphological analysis of individual cells as we employed blind in vivo recording methods (*Figure 1*). We thus focused on the visual response properties for categorizing cell classes in the following two ways.

First, we classified cells in vivo using the response quality, response polarity, and motion sensitivity (*Figure 2*). Specifically, we first divided the cells into two groups: reliably responsive ones with SNR > 0.15 (*Equation 2*) and the other low-quality ones ('N/A' type; *Figure 2B*). We then set thresholds at ±0.25 for the ON–OFF index (*Equation 3*) to identify ON, OFF, and ON/OFF cells within the reliably responsive cells that increased firing in response to light increments, light decrements, and both, respectively (*Figure 2C*). For some cells where these measures were not available, we used the responses to the full-field sinusoidally flickering stimuli to calculate the response quality and polarity in a similar manner. Within the ON cells, we further identified an 'OFF-suppressive' class if they had a significant negative amplitude in *Equation 6* with 95% confidence intervals in the parameter estimation. Independently, we also labeled cells as motion sensitive or not, based on the DS/OS indices as described above (*Equation 3*; *Figure 2D–F*).

Second, because these measures are not available for the ex vivo data sets (*Vlasiuk and Asari, 2021*), we performed classification using the temporal filter properties to make a fair comparison between the ex vivo and in vivo data sets (*Figures 5 and 6*). Specifically, we first ran a PCA on the temporal filters (*L2*-normalized STA) obtained in each condition (*Gollisch and Meister, 2008*; *Asari and Meister, 2014*). The first two principal components (PC1, monophasic filter; and PC2, biphasic filter) were largely sufficient to fit all the temporal filter dynamics, accounting for 78–86% of the total variance in each condition. Each temporal filter is a point in the PCA biplot (i.e., the two-dimensional space spanned by PC1 and PC2), and we grouped its shape into four subtypes based on its position: monophasic OFF, biphasic OFF, monophasic ON, and biphasic ON from the first to the fourth quadrants, respectively.

Here we did not aim to fully differentiate all functional cell types (~35 RGC types from ex vivo studies; *Baden et al., 2016*; *Jouty et al., 2018*) for three reasons. First, our data sets are not large enough. Second, the presented stimulus sets were not diverse enough. Finally, some cells may have not been sampled, such as those projecting to the ipsilateral side of the brain, or those projecting exclusively via the retinohypothalamic tract (e.g., to the medial terminal nucleus of the accessory optic system; *Yonehara et al., 2009*; *Gauvain and Murphy, 2015*). Thus, here we did not use the term 'cell type' to avoid confusion.

## Behavior data analysis

We monitored the behavioral states of an animal during in vivo recordings in two ways (*Figure 1—figure supplement 1*). First, we monitored the running speed based on the turning speed of the

custom-made rotary treadmill. Second, we videotaped the animal's eye, facing toward the visual stimulation screen. Eye tracking was done by passing individual frames of the eye videos to Mask R-CNN, an object detection and segmentation neural network (*He et al., 2017*; *Abdulla, 2017*). Mask R-CNN was fine-tuned with 500 manually annotated frames of our recordings, covering the range of pupil size and background illumination found in our setup. The detected masks were fitted with an ellipse, retrieving the center coordinate, semi-major and -minor axes, and rotation angle (e.g., *Figure 1—figure supplement 1A*).

Pairwise cross-correlation (100 ms bin) was calculated between the RGC firing rate, running speed and pupil size (*Figure 2—figure supplement 1*) for the recording period where the mean light intensity was relatively stable: that is, randomly flickering full-field stimulation and moving grating stimulation periods. The peak value was then selected within 5 s lag while the p-value was calculated from the data outside the range. The peak cross-correlation was similarly calculated between the pupil size and the light intensity for the full-field sinusoidally flickering stimulus and the full-field contrast inversion stimulation periods.

## Model analysis

### Information theoretic analysis

We took an information theoretic approach to quantify the amount of information carried by the retinal output spike trains (*Figure 7*; *Rieke et al., 1997*), focusing on the randomly flickering full-field stimulation period ($T$ = 300 s). Specifically, we calculated the information conveyed by a single spike, $I_{\text{spike}}$ bits/spike, using the following formula:

$$I_{\text{spike}} = \frac{1}{T} \int_0^T dt \left[ \frac{r(t)}{\bar{r}} \right] \log_2 \left[ \frac{r(t)}{\bar{r}} \right], \tag{8}$$

where $r(t)$ is the instantaneous firing rate of a target cell using small time bins $\Delta t$, and $\bar{r}$ is the average firing rate over time. To avoid the estimation bias due to the finite data size, we computed $I_{\text{spike}}$ in *Equation 8* as a function of bin size $\Delta t$, performed a linear regression to the well-behaved linear portion of these measurements, and extrapolated to the limit $\Delta t \to 0$ (i.e. the $y$-intercept). The information rate conveyed by a target cell was then given as: $I_{\text{rate}} = \bar{r} \cdot I_{\text{spike}}$ bits/s.

### Rate and latency coding models

We took a modeling approach to examine how retinal coding is affected by the response characteristics observed under different recording conditions (*Figure 7—figure supplement 1*; *Gollisch and Meister, 2008*). Specifically, using the temporal filter (STA; *Figure 5*) and static nonlinear gain function (*Equation 8*; *Figure 6*) in the linear–nonlinear cascade model framework, we first simulated the responses of each cell to a step change in light intensity at 10 different levels of contrast increase and decrease, respectively ($n_{\text{level}}$ = 20 in total; e.g., *Figure 7—figure supplement 1A and B*). To estimate the information contained in a given characteristic $x$ of the simulated responses, we then used the following formula:

$$I = \sum_s \int dx \, p(s) \, p(x \mid s) \, \log_2 \frac{p(x \mid s)}{p(x)}, \tag{9}$$

where $p(s) = 1/n_{\text{level}}$ is the probability of the stimulus $s$; $p(x \mid s)$ is the conditional probability density of $x$ given stimulus $s$; and $p(x) = \sum_s p(s) \, p(x \mid s)$ is the total probability density of $x$, obtained as the normalized sum of $p(x \mid s)$. For rate coding (*Figure 7—figure supplement 1E*), we focused on the peak firing rate of the simulated responses, and assumed that $p(x = \text{"peak rate"} \mid s)$ follows a Gaussian distribution with the mean and standard deviation of the peak firing rate and its 2.5% value, respectively (e.g., *Figure 7—figure supplement 1C*). For temporal coding (*Figure 7—figure supplement 1F*), we focused on the response latency, where we assumed that the detection threshold was given by 10 ± 2.5% deviation from the baseline firing rate (mean ± standard deviation of a Gaussian distribution), from which $p(x = \text{"latency"} \mid s)$ was calculated (e.g., *Figure 7—figure supplement 1D*). For stimuli that did not trigger strong enough responses to cross the detection threshold, the response latency was considered to be infinity.

## Linear decoding model

We used a linear decoding model to investigate how population coding of the retina depends on the recording conditions (*Figure 8*). Specifically, we first computed the optimal linear decoder (333 ms window with a bin size of 16.7 ms) from the simultaneously recorded population activity in response to the full-field randomly flickering stimulus (*Warland et al., 1997*). The performance of the linear decoder was measured by Pearson's correlation coefficient between the presented and estimated stimuli with 10-fold cross-validation. For each recording condition, the decoding performance as a function of the number of input cells, $n$, was then assessed by fitting the following exponential saturation function with a plateau at 1:

$$f(n) = 1 - (1 - a)\, e^{-(n-1)/b}, \tag{10}$$

where $a = f(1) \in [0, 1]$ represents the single-cell decoding performance; and $b > 0$ is the saturation constant, hence a larger value of $b$ indicates that the linear decoding requires a larger population to improve its performance.

## Acknowledgements

This work was supported by research grants from EMBL (HA). The EMBL Histology Facility and the Advanced Light Microscopy Facility are acknowledged for support in sample preparation and image acquisition for histological analyses, respectively. EMBL IT Support is acknowledged for provision of computer and data storage servers; and the LAR facility for taking care of animals. We thank Dmitry Molotkov for his help in setting up the recording rig, and all the Asari lab members as well as Cornelius Gross and Santiago Rompani for many useful discussions.

## Additional information

### Funding

| Funder | Grant reference number | Author |
| --- | --- | --- |
| European Molecular Biology Laboratory | | Hiroki Asari |

The funders had no role in study design, data collection, and interpretation, or the decision to submit the work for publication.

### Author contributions

Tom Boissonnet, Conceptualization, Data curation, Formal analysis, Investigation, Writing – original draft, Writing – review and editing; Matteo Tripodi, Data curation, Formal analysis, Investigation, Writing – review and editing; Hiroki Asari, Conceptualization, Formal analysis, Supervision, Funding acquisition, Writing – original draft, Writing – review and editing

### Author ORCIDs

Tom Boissonnet ⓘ https://orcid.org/0000-0002-3328-9467
Hiroki Asari ⓘ http://orcid.org/0000-0003-3396-1935

### Ethics

All experiments were performed under the license 233/2017-PR from the Italian Ministry of Health, following protocols approved by the Institutional Animal Care and Use Committee at European Molecular Biology Laboratory.

### Decision letter and Author response

Decision letter https://doi.org/10.7554/eLife.78005.sa1
Author response https://doi.org/10.7554/eLife.78005.sa2

## Additional files

### Supplementary files
• MDAR checklist

### Data availability
All data and codes used in this study are available on Dryad.

The following dataset was generated:

| Author(s) | Year | Dataset title | Dataset URL | Database and Identifier |
|-----------|------|---------------|-------------|-------------------------|
| Boissonnet T, Tripodi M, Asari H | 2023 | Data from: Awake responses suggest inefficient dense coding in the mouse retina | https://doi.org/10.5061/dryad.5x69p8d52 | Dryad Digital Repository, 10.5061/dryad.5x69p8d52 |

The following previously published dataset was used:

| Author(s) | Year | Dataset title | Dataset URL | Database and Identifier |
|-----------|------|---------------|-------------|-------------------------|
| Vlasiuk A, Asari H | 2021 | Feedback from retinal ganglion cells to the inner retina | https://doi.org/10.5281/zenodo.5057577 | Zenodo, 10.5281/zenodo.5057577 |

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
