## [Editor Report]

This paper compares output signals from the mouse retina in three conditions: awake mice, anesthetized mice, and isolated retinas. The paper provides compelling evidence for substantial and important differences, particularly between awake and either of the other conditions. Retinal signaling has been well studied using ex vivo preparations, with an assumption that the findings from those studies can be carried over to how the retina operates in vivo. The results from this paper at a minimum indicate a need to be cautious about that assumption.

---

## [Decision Letter]

**Decision letter after peer review:**

Thank you for submitting your article "Awake responses suggest inefficient dense coding in the mouse retina" for consideration by *eLife*. Your article has been reviewed by 3 peer reviewers, one of whom is a member of our Board of Reviewing Editors, and the evaluation has been overseen by Ronald Calabrese as the Senior Editor. The following individual involved in review of your submission has agreed to reveal their identity: Gregory William Schwartz (Reviewer #3).

Essential revisions:

All three reviewers were enthusiastic about the work described in the paper and highlighted the uniqueness and importance of the data. The consultation among the reviewers also served to emphasize several key points that need to be strengthened. This is not a complete list of the needed changes (see the individual reviews for more details) but instead highlights several issues shared across reviewers.

1) Identification of cell types. Recorded RGCs were separated into broad categories but not into the defined cell types known to be present in mouse. The wording in the paper in several places seems to suggest that cell types were identified. This includes the suggestion that the high maintained firing rate On cells might be a new cell type. The language surrounding the correspondence between the recorded RGCs and known RGC types should reflect the limitations of the current approaches more clearly. Related to this point, the number of cells recorded in the awake state is relatively small given that there are ~35 RGC types (i.e. some may not have been sampled at all).

2) Eye movements and behavioral state. A distinguishing feature of the recordings from awake mice is that the animals are behaving. Eye movements are considered as a possible explanation for the lack of directionally-selective responses, but could eye movements also modulate the input to other RGC types and hence account for some of the differences in responses? More information about pupil size, eye movements and behavioral state during the recordings would help evaluate the differences in response observed.

3) Connection with the existing literature. The paper (rightfully) points out that there is little information about RGC responses in awake animals. But there is some, and those past studies should be summarized and compared to the present results.

4) Relationship with efficient coding ideas. Several other lines of evidence suggest that efficient coding cannot alone account for RGC response properties. The efficient coding arguments in the paper need to be related to that past work.

5) Spike sorting and light levels. More details are needed on both of these technical points, as detailed in the individual reviews.

*Reviewer #1 (Recommendations for the authors):*

Line 82: This sentence – and later one – does not appear correct. Although the On cells retain their polarity, they are responding to a different range of intensities.

Line 107: I think most people will read "RGC types" as more specific than what is meant here. Perhaps "classes" or just stick to On, Off and On/Off.

Line 124-127: Sentence is awkward.

Line 149: Sentence is awkward.

Line 379: It is less clear that a difference in effective threshold can account for differences in temporal tuning.

*Reviewer #2 (Recommendations for the authors):*

– General: for the interpretation of the awake data, it would be important to know the details about the behavioral state (as indexed e.g. by running, pupil size) of the animal and any eye movements.

– l. 42: I suggest to add one more sentence clarifying that the previous results have mostly been obtained under anesthesia.

– l. 119: Can the authors back up this speculation by analysing eye movements in the awake and anesthetized condition?

– l. 125: The authors should present data showing that their recordings are better in resolving RGC types and that they are able to cluster responses into the 30 types observed ex vivo or revise the sentence. Along these lines, given the recording location within the OT, could you infer how many different types should still be present in the sample?

– Figure 3: add a box around the white part of the stimulus.

– l. 158: Do the authors have any evidence that firing rate is indeed near saturation?

– l. 176 ff: Please rephrase to better indicate whether the rates are rates relative to baseline B or absolute rates; please also describe more details regarding the clustering of response phases.

– l. 176 ff: Regarding the changes in contrast response functions between awake and anaesthetised recordings, in my view, the authors should corroborate their conclusions regarding the dynamic range of RGCs with a consideration of the steepness of the sigmoid. Also, can they extend their conclusions about dynamic range to the population level, e.g. investigating the variability of the midpoint across RGCs, similar to Keller et al. (2015) J Neurosci? I would also recommend using terms like "contrast sensitivity" in accordance with the literature – the authors use it here to describe the response magnitude at 10% contrast, which is not a common definition of contrast sensitivity. For the awake conditions, it seems like the basic division in ON and OFF RGCs does not yield distinct response phases. Could the authors please provide more details on what is happening here?

– Figure 4: Can the authors please provide evidence that fit quality is an appropriate measure of responsiveness? There is a wide range of fit qualities, so one idea could be to show how a poor fit corresponds to poor responsiveness. Also, how do the results differ across cell types?

– Figure 5: With regards to the comparison to previous ex vivo results, I would advise the authors to make the comparison of the visual stimulation conditions, in particular the luminance regime, between ex vivo and in vivo conditions more explicit. It is important to rule out low-level difference, e.g. in luminance regimes between ex vivo conditions, where light can directly hit the photoreceptors, and the in vivo situation presented here, where light is being project via a mirror to a screen, from which it is reflected and has to travel through the lens etc. before impinging on the photoreceptors. The authors should also clarify why they have a maximum light intensity of 36 mW/m^2 on the isolated retina in the methods section, but talk about 18 mW/m^2 in the results.

---

## [Author Response]

Essential revisions:All three reviewers were enthusiastic about the work described in the paper and highlighted the uniqueness and importance of the data. The consultation among the reviewers also served to emphasize several key points that need to be strengthened. This is not a complete list of the needed changes (see the individual reviews for more details) but instead highlights several issues shared across reviewers.1) Identification of cell types. Recorded RGCs were separated into broad categories but not into the defined cell types known to be present in mouse. The wording in the paper in several places seems to suggest that cell types were identified. This includes the suggestion that the high maintained firing rate On cells might be a new cell type. The language surrounding the correspondence between the recorded RGCs and known RGC types should reflect the limitations of the current approaches more clearly. Related to this point, the number of cells recorded in the awake state is relatively small given that there are ~35 RGC types (i.e. some may not have been sampled at all).

We made substantially more recordings in the awake condition, reaching 282 cells (in 15 animals) in total. With the new data sets combined, we categorized the responses into 5 major classes (ON, OFF-suppressive ON, ON/OFF, OFF, and the rest), each subdivided into direction-selective, orientation-selective, and non-selective groups. We did not further subdivide the cell classes because 1) our data sets are not large enough yet; 2) the presented stimulus sets were not diverse enough to fully differentiate functional cell-types (~35 RGC types from ex vivo studies; Baden et al., 2016; Jouty et al., 2018); and 3) some cell-types may have not been sampled, such as those projecting to the ipsilateral side of the brain, or those projecting exclusively via the retinohypothalamic tract (e.g., to the medial terminal nucleus of the accessory optic system; Yonehara et al., 2009; Gauvain and Murphy, 2015).

Note however that it is technically difficult to collect data from thousands of cells as in ex vivo studies since we can record only a dozen of cells at a time with a limited recording length. Moreover, awake recordings generally have a higher noise level, much as in recordings from other brain areas (Durand et al. 2016; De Franceschi and Solomon, 2018). In particular, it is not trivial to compensate for the eye motion due to its nonlinear transformation (though infrequent in the head-fixed condition; Samonds et al. 2018; Miura and Scanziani, 2022), and this is why we mainly used full-field stimulation in this study, except for moving gratings. Despite those limitations, we think we have collected enough datasets to capture the basic properties of the retinal output in vivo, clarifying the robust differences between awake and anesthetized responses.

The revised manuscript clarified these points as well as the technical limitations, and avoided using the term “types” as suggested by the reviewers.

2) Eye movements and behavioral state. A distinguishing feature of the recordings from awake mice is that the animals are behaving. Eye movements are considered as a possible explanation for the lack of directionally-selective responses, but could eye movements also modulate the input to other RGC types and hence account for some of the differences in responses? More information about pupil size, eye movements and behavioral state during the recordings would help evaluate the differences in response observed.

We videotaped the eye movement of the subject animals during recordings. For the original datasets, however, we were not able to reliably analyze the videos due to intermittent frame drops. For the newly acquired datasets, we fixed the technical problem to examine the effects of eye movements, pupil dynamics, as well as running behaviour on the retinal output in vivo as suggested by the reviewers (e.g., Figure 1 —figure supplement 1A-G).

1) We first examined the basic statistics of those parameters (Figure 1 —figure supplement 1H-J). In the awake condition, we found a certain bias in the direction of the eye movements: i.e., primarily in the horizontal directions and much less in the vertical directions (Figure 1 —figure supplement 1H; see also Samonds et al. 2018; Miura and Scanziani, 2022). This is contradictory to our previous reasoning for the general lack of directionally-selective (DS) responses. Furthermore, the spatial frequencies of our stimulus (square waves at 3° or 20°) was not optimal to drive optokinetic responses (Samonds et al. 2018); and the eye movements cannot explain the lack of orientationally-selective (OS) responses. We thus removed the corresponding sentences in the revised manuscript.

We now instead think that the lack of DS/OS responses in the awake condition is because of the high firing activity. The DS/OS indices were calculated as a normalized population vector (Equation (3); Masurek et al., 2014). Thus, a large denominator in awake responses inevitably led to a small DS/OS index value. We indeed found a negative correlation between the average firing rate during the stimulus presentation period and the DS/OS indices as shown in Author response image 1:

**Author response image 1. sa2fig1:** 

Note that, under anesthesia, no eye or body movements were observed besides a small and slow drift of the pupil size as shown in Author response image 2 for a representative case (probability density of pupil center location (A), pupil size (B), and running speed (C) across different visual stimulation periods; same format as in Figure 1 —figure supplement 1H-J):

We next ran a series of correlation analyses. Firstly, pupil size was negatively correlated with the overall light intensity of the visual stimulus (-0.18 ± 0.07; N=19 animals) due to pupillary reflexes in awake animals (Hatter et al., 2003). This does not fully compensate for the light intensity changes, and thus the amount of photons passing through the pupil is still higher at a higher light intensity, driving ON cells and suppressing OFF cells. Nevertheless, this means that the dynamic range of the light intensity on the retina was smaller than was intended in awake mice. Given that most RGCs show larger responses to higher contrast stimuli (Figure 3; see also Wang et al., 2011; Baden et al. 2016), one would then expect smaller responses in the awake condition than in the anesthetized condition where pupillary reflexes were suppressed. This was, however, not the case: awake responses were generally larger (Figure 3) despite the counteracting effect of the pupillary reflexes.Secondly, when the average light intensity was stable over time, pupil size was correlated with RGC firing in a response-polarity-dependent manner (Figure 2 —figure supplement 1A,B; Pearson’s *R*=0.38, *p*<0.001). It has been suggested that pupil size is modulated by attentional states (Hoeks and Levelt, 1993). Attentional modulation of visual processing (Reynolds and Chelazzi, 2004) may then be explained partly by the changes in the retinal activity due to changes in the amount of light impinging on the retina upon changes in the pupil size.

Thirdly, as reported previously, pupil size was positively correlated with locomotion (0.34 ± 0.20, N=19; see also Eriksen et al., 2014), and locomotion was in turn correlated with the firing activity of some RGCs in either polarity (Figure 2 —figure supplement 1C,D; see also Schroeder et al., 2020). The locomotion effects were not dependent on the visual response polarity of the cells (Figure 2 —figure supplement 1D; Pearson’s *R*=0.11, p=0.3). Thus the correlation between locomotion and RGC firing dynamics cannot be simply explained by the light intensity fluctuation due to the associated pupil dilation.

The diagram in Author response image3 summarizes the correlation analysis results.

**Author response image 3. sa2fig3:** 

The frequency of the eye movements (saccades and blinks) during the head-fixed awake recordings was very low (0.06 +/- 0.03 Hz; mean +/- standard deviation, N=19 animals; see Figure 1 —figure supplement 1 for example), consistent with past studies (Samonds et al. 2018; Miura and Scanziani, 2022). Thus, their contribution to our overall conclusion was considered negligible. Indeed, the response features remained the same in the reversecorrelation analysis even after excluding all the eye movement periods (-100 ms to +400 ms window around the event), as shown in Author response image 4:

**Author response image 4. sa2fig4:** 

In sum, awake responses had different characteristics from anesthetized or ex vivo responses even after eliminating the effects of confounding factors, such as eye movements and behavioural states. These effects are for sure an interesting topic for future studies. The revised manuscript clarified the above points.

3) Connection with the existing literature. The paper (rightfully) points out that there is little information about RGC responses in awake animals. But there is some, and those past studies should be summarized and compared to the present results.

To the best of our knowledge, spiking activity of RGCs in awake animals has been reported only in some recent studies with a low sample size (Hong et al., 2018; Schroeder et al., 2020; Sibille et al., 2022). These studies did not provide any descriptive statistics on the awake RGC response properties, hence making it difficult to make a direct comparison to our study. Nevertheless, the reported examples in Figure 3C of Schroeder et al. (2020) and in Figure S7h of Sibille et al. (2022) show high baseline activity (>10 Hz), consistent with our findings. These two studies employed high-density electrodes (NeuroPixels) to record axonal signals of RGCs in C57BL/6J mice, and thus are arguably comparable to our study from a methodological viewpoint as well, although visual stimuli were presented from an LCD screen at 1-2 orders of magnitude higher light intensity. In contrast, using mesh electrodes directly injected into the mouse eye, Hong et al. (2018) reported relatively low light-evoked responses (<10 Hz) in albino CD1 strain. This is likely not due to the experimental hardware employed, but due to the mouse strain. Albino mice generally have poor vision with a reduced number of rod photoreceptors (Abdeljalil et al. 2005; Brown et al. 2007), suggesting a certain level of deficits in their retina. It is an interesting topic for future studies to further characterize the awake RGC activities in various conditions across different strains/species.

We clarified the point in the revised discussion.

4) Relationship with efficient coding ideas. Several other lines of evidence suggest that efficient coding cannot alone account for RGC response properties. The efficient coding arguments in the paper need to be related to that past work.

We totally agree that we are not the first to argue against the efficient coding theory in the retina (e.g., Schwartz, 2021). Our main argument is that certain aspects of the RGC activity are distinct in an awake condition, such as the baseline firing and response kinetics, and thus we cannot simply translate our knowledge obtained from ex vivo studies into awake animals. The efficient coding framework was employed here as an example to highlight such differences. In an isolated retina, on the one hand, while some RGCs exhibit high firing under certain conditions (Ke et al. 2014; Smeds et al. 2019), many RGCs fire only sparsely as expected from the efficient coding principles (Pitkow & Meister, 2012; Doi et al., 2012). In an awake condition, on the other hand, we found a general increase of the firing (Figures 3 and 6). To further explore the implication of this finding on retinal computations, we showed in the revised manuscript that 1) awake responses had a comparable total information transfer rate (in bits per second; Figure 7A) but were generally less efficient (i.e., lower bits per spike; Figure 7B); 2) awake responses were in favor of a rate code but not a latency-based temporal code (Figure 7 —figure supplement 1); and 3) a linear decoder worked significantly better with the population responses in awake mice (Figure 8), even though an image reconstruction is not necessarily the objective function of the retina or the visual system. These results point out a need to reconsider retinal function in vivo, including the efficient coding theory.

We revised the texts in the Discussion section accordingly.

5) Spike sorting and light levels. More details are needed on both of these technical points, as detailed in the individual reviews.

The revised manuscript clarified the technical details, including spike sorting and light levels. In short, we provided the procedure and key parameter values for spike sorting (SpykingCircus, Yger et al., 2018; Phy, Rossant, 2020) in the revised method section, and showed spike waveforms and autocorrelograms for three representative units in the revised Figure 1. The revised methods also provided an estimated photoisomerization rate in each recording condition (on the order of 10^3^ and 10^4^ R*/s/photoreceptor for in vivo and ex vivo recordings, respectively). For details, see also our response to the individual reviews below.

Reviewer #1 (Recommendations for the authors):Line 82: This sentence – and later one – does not appear correct. Although the On cells retain their polarity, they are responding to a different range of intensities.

In the revised manuscript, we quantified the amount of information conveyed and showed that awake responses had a comparable total information transfer rate (in bits per second) but were generally less efficient, with lower bits per spike, (Figure 7) because of the high baseline activity (Figure 3).

Line 107: I think most people will read "RGC types" as more specific than what is meant here. Perhaps "classes" or just stick to On, Off and On/Off.

Following the reviewer’s suggestion, we avoided the term “type” in the revised manuscript.

Line 124-127: Sentence is awkward.

We removed the sentence because here we cannot evaluate the recording techniques *per se*.

Line 149: Sentence is awkward.

We corrected the sentence as follows in the revised manuscript: “We fitted to the response an even-power of the sine function, weighted with a sigmoid envelope (Equation (6) in Methods).”

Line 379: It is less clear that a difference in effective threshold can account for differences in temporal tuning.

We do not think that the spike threshold changes in different conditions. Instead, awake cells have a shorter response latency likely because of higher conductance as suggested by the higher baseline activity. While the exact mechanisms are unknown, this is consistent with the observations in the downstream visual areas, including LGN and V1 (Wang et al. 2014; Durand et al. 2016) as well as in the auditory system (Zohar et al., 2011). We clarified the point in the revised manuscript.

Reviewer #2 (Recommendations for the authors):– General: for the interpretation of the awake data, it would be important to know the details about the behavioral state (as indexed e.g. by running, pupil size) of the animal and any eye movements.

Following the reviewer’s suggestions, we examined the behavioural data during the recordings as well. For example, we ran a series of correlation analyses to explore the modulation of RGC firing in awake animals (Figure 1 —figure supplement 1; Figure 2 —figure supplement 1). For details, see above our responses to the Essential Review item #2. In short, while pupil size was positively correlated with the visual response polarity of RGCs (Figure 2 —figure supplement 1A,B), it was negatively correlated with the light intensity due to pupillary reflexes, making the dynamic range smaller than intended. Thus, the general increase of RGC firing in awake animals cannot be explained by the pupil size modulation because most RGCs show larger responses to higher contrast stimuli (Figure 3; see also Wang et al., 2011; Baden et al. 2016). Pupil size was also positively correlated with running speed, while running speed was either positively or negatively correlated with RGC firing (Figure 2 —figure supplement 1C,D). The polarity of the locomotion effects was not significantly dependent on the visual response polarity of the cells. The effects of the eye movements were negligible as they occurred infrequently in the head-fixed condition (Figure 1 —figure supplement 1; Samonds et al. 2018; Miura and Scanziani, 2022).

We have clarified these points in the revised manuscript.

– l. 42: I suggest to add one more sentence clarifying that the previous results have mostly been obtained under anesthesia.

Thanks for the suggestion. We have clarified the point in the revised manuscript.

– l. 119: Can the authors back up this speculation by analysing eye movements in the awake and anesthetized condition?

We have revised the text here because the data did not support our original speculation. In short, a bias in the direction of the eye movements (Figure 1 —figure supplement 1H) and their infrequency in head-fixed mice do not explain a general reduction of DS/OS indices in awake animals. We now instead think that this is due to the high baseline activity. For details, see our response to the Essential Review item #2. Of note, no eye or body movements were observed under anesthesia besides a small and slow drift of the pupil size.

– l. 125: The authors should present data showing that their recordings are better in resolving RGC types and that they are able to cluster responses into the 30 types observed ex vivo or revise the sentence. Along these lines, given the recording location within the OT, could you infer how many different types should still be present in the sample?

We have removed the sentence in the revised manuscript. The visual responses were classified only into broad categories, but not into the full repatours as were identified in the past ex vivo studies. We apologize for the confusion.

Given that most RGCs project their axons through the optic tract (Ellis et al., 2016; Rosón et al., 2019), we expect that the optic tract recordings should have access to most cell types, if not all. However, a larger data set with a more diverse set of visual stimuli will be required to perform better cell-type classifications in a more systematic manner than what we obtained in this study. This is for sure an interesting and important direction for future studies.

We clarified these points in the revised manuscript.

– Figure 3: add a box around the white part of the stimulus.

We followed the reviewer’s suggestion in the revised figure.

– l. 158: Do the authors have any evidence that firing rate is indeed near saturation?

No, we do not. Here we observed that the “OFF-suppressive” ON cells had higher baseline firing than the other classes of cells (~60 Hz; Figure 3D); and that they showed a prominent decrease of the firing in response to OFF stimuli, but no further increase of the firing in response to slow increases in light intensity (ON stimuli; e.g., Figure 3B). We thus speculated that their firing was near saturated. However, we have not directly examined saturation. To avoid confusion, we thus removed the phrase “near saturation” in the revised manuscript.

– l. 176 ff: Please rephrase to better indicate whether the rates are rates relative to baseline B or absolute rates; please also describe more details regarding the clustering of response phases.

Here we focused on the evoked firing rate relative to baseline (i.e., the parameter *A* in Equation (6) in Methods). We used the K-means algorithm to cluster the response phase (into ON and OFF). We clarified these points and gave more details in the revised manuscript.

– l. 176 ff: Regarding the changes in contrast response functions between awake and anaesthetised recordings, in my view, the authors should corroborate their conclusions regarding the dynamic range of RGCs with a consideration of the steepness of the sigmoid.

Thanks for the suggestion. Author response table 1 is the summary table of the parameters for contrast response function (“sigmoid”; Equation (6) in Methods) across different in vivo recording conditions (median ± interquartile range). All the evoked response values are relative to the baseline.

**Author response table 1. sa2table1:** 

	Awake	FMM	Isoflurane	P-value (Kruskal-Wallis test)
	(N=247)	(N=95)	(N=147)	
Baseline B (Hz)	20±27	2±4	3±10	*P*<0.001
Evoked absolute amplitude |A| (Hz)	45±55	27±33	30±31	*P*<0.001
Midpoint of sigmoid t_0_ (% contrast)	70±61	45±44	45±32	*P*<0.001
Steepness of sigmoid λ (/s)	0.44±0.53	0.67±0.79	0.69±0.62	*P*<0.001
Evoked response at 10% contrast (Hz)	7±13	3±10	4±12	*P*<0.001
Slope at midpoint t_0_ (Hz/% contrast)	5±11	5±7	6±9	*P*=0.1

We found several distinct features in the awake responses in comparison to those under anesthesia.

RGCs of awake mice had a higher baseline firing (B, “lower” bound of sigmoid) than under anesthesia, consistent with the past observation in LGN and V1 (Wang et al. 2014; Durand et al. 2016).

Awake RGC responses had a larger dynamic range, with a higher evoked response amplitude (|A|, “upper” bound of sigmoid), a higher midpoint of sigmoid (t_0_), and a lower steepness of sigmoid (λ) than the anesthetized responses.

Awake responses showed a higher sensitivity at low contrast, as indicated by a higher evoked response at 10% contrast. The slope size at the midpoint of sigmoid (|Aλ/4|) was comparable across different conditions, suggesting that the cell’s maximum sensitivity to a change in contrast was not significantly different across conditions.

We clarified these points in the revised manuscript.

Also, can they extend their conclusions about dynamic range to the population level, e.g. investigating the variability of the midpoint across RGCs, similar to Keller et al. (2015) J Neurosci?

Following the reviewer’s suggestion, we further analyzed the contrast response function properties across different response polarities: ON (including OFF-suppressive ON), OFF, or ON/OFF. We found that the baseline firing rate was higher for ON/OFF cells than for ON or OFF cells (p<0.001; Kruskal-Wallis test). Otherwise we did not find any significant dependence on the response polarity at the population level (see Author response table 2, summary table).

Author response table 2.

I would also recommend using terms like "contrast sensitivity" in accordance with the literature – the authors use it here to describe the response magnitude at 10% contrast, which is not a common definition of contrast sensitivity.

To avoid confusion, we replaced the phrase “contrast sensitivity” to a neutral phrase, such as “the sensitivity to contrast” in the revised manuscript.

For the awake conditions, it seems like the basic division in ON and OFF RGCs does not yield distinct response phases. Could the authors please provide more details on what is happening here?

This was due to a scarcity of data in the original manuscript. With a larger dataset we collected for the revision (Figure 3F), the response phase was largely and more clearly divided into ON and OFF cells in all recording conditions, while ON/OFF cells can be in either cluster. Note that the response phase was significantly shifted towards smaller values in the awake condition than in the anesthetized conditions, indicating faster response kinetics in the awake condition. This is consistent with the analysis results of the temporal frequency sensitivity (Figure 4) and STA (Figure 5).

– Figure 4: Can the authors please provide evidence that fit quality is an appropriate measure of responsiveness? There is a wide range of fit qualities, so one idea could be to show how a poor fit corresponds to poor responsiveness. Also, how do the results differ across cell types?

To address this question, we have examined the power spectral density of the responses at the given temporal frequency of the full-field flickering stimulus (see Author response image 4). We generally found 1) a low power for cells with a low fit quality (below threshold, 0.2; grey); and 2) a positive correlation for those with a high fit quality (above threshold; color-coded as in Figure 4). This supports that fit quality (Equation (5) in Methods) is a reasonable measure of a cell’s responsiveness.

Across the response polarities, we found that ON/OFF cells (N=170) were most responsive at a medium frequency (7.5Hz), while ON or OFF cells (N=199 and 121, respectively) at low-tomedium frequencies (1.875 – 7.5 Hz). Indeed, the fraction of responsive cells under our criteria was significantly lower for ON/OFF cells than for ON or OFF cells at low frequencies (1.875 and 7.5 Hz; p<0.001, two-proportion z-test) as shown in Author response image 5:

**Author response image 5. sa2fig5:** 

– Figure 5: With regards to the comparison to previous ex vivo results, I would advise the authors to make the comparison of the visual stimulation conditions, in particular the luminance regime, between ex vivo and in vivo conditions more explicit. It is important to rule out low-level difference, e.g. in luminance regimes between ex vivo conditions, where light can directly hit the photoreceptors, and the in vivo situation presented here, where light is being project via a mirror to a screen, from which it is reflected and has to travel through the lens etc. before impinging on the photoreceptors. The authors should also clarify why they have a maximum light intensity of 36 mW/m^2 on the isolated retina in the methods section, but talk about 18 mW/m^2 in the results.

We apologize for the confusion. In the original manuscript, the number in the Results section was the average light intensity (or “gray”), while the number in the methods section was the maximum light intensity (or “white”). To avoid confusion, we described only the maximum light intensity throughout the revised manuscript, and calculated the photoisomerization rate for further clarification.

As detailed in our response to the reviewer #1 (“Light Levels” section), the estimated photoisomerization rate was about 10 times larger for the ex vivo recordings (~10^4^ R*/photoreceptor/s) than for the in vivo recordings (~10^3^ R*/photoreceptor/s). Both recordings were thus made in a high mesopic / low photopic condition where both rods and cones are active. Because RGCs typically show smaller and slower responses to lower contrast stimuli (Wang et al., 2011; Tikidji-Hamburyan et al. 2015; Borghuis et al., 2018), this light intensity difference between ex vivo and in vivo setups cannot account for our observations that awake responses had higher baseline firing and fast kinetics. It should also be noted that all in vivo recordings were done in the same experimental set up besides anesthesia.